# Secondary structural ensembles of the SARS-CoV-2 RNA genome in infected cells

Tammy C. T. Lan[1,2], Matty F. Allan[1,3,4,5], Lauren E. Malsick [6], Jia Z. Woo[1,3], Chi Zhu[7,8], Fengrui Zhang[1], Stuti Khandwala[1,9,10], Sherry S. Y. Nyeo[1,9,10], Yu Sun[11], Junjie U. Guo [11], Mark Bathe[4], Anders Näär[7,8], Anthony Griffiths [6] & Silvi Rouskin [1,3✉]

SARS-CoV-2 is a betacoronavirus with a single-stranded, positive-sense, 30-kilobase RNA genome responsible for the ongoing COVID-19 pandemic. Although population average structure models of the genome were recently reported, there is little experimental data on native structural ensembles, and most structures lack functional characterization. Here we report secondary structure heterogeneity of the entire SARS-CoV-2 genome in two lines of infected cells at single nucleotide resolution. Our results reveal alternative RNA conformations across the genome and at the critical frameshifting stimulation element (FSE) that are drastically different from prevailing population average models. Importantly, we find that this structural ensemble promotes frameshifting rates much higher than the canonical minimal FSE and similar to ribosome profiling studies. Our results highlight the value of studying RNA in its full length and cellular context. The genomic structures detailed here lay groundwork for coronavirus RNA biology and will guide the design of SARS-CoV-2 RNA-based therapeutics.

[1] Whitehead Institute for Biomedical Research, Cambridge, MA, USA. [2] Department of Molecular and Cellular Biology, Harvard University, Cambridge, MA, USA. [3] Department of Microbiology, Harvard Medical School, Boston, MA, USA. [4] Department of Biological Engineering, Massachusetts Institute of Technology, Cambridge, MA, USA. [5] Computational and Systems Biology, Massachusetts Institute of Technology, Cambridge, MA, USA. [6] National Emerging Infectious Diseases Laboratories, Boston University School of Medicine, Boston University, Boston, MA, USA. [7] Department of Nutritional Sciences & Toxicology, University of California, Berkley, CA 94720, USA. [8] Innovative Genomics Institute, University of California, Berkley, CA 94720, USA. [9] Department of Biology, Massachusetts Institute of Technology, Cambridge, MA, USA. [10] Department of Electrical Engineering & Computer Science, Massachusetts Institute of Technology, Cambridge, MA, USA. [11] Department of Neuroscience, Yale University School of Medicine, New Haven, CT, USA. ✉email: silvi_rouskin@hms.harvard.edu

Severe acute respiratory syndrome coronavirus 2 (SARS-CoV-2), the causative agent of coronavirus disease 2019 (COVID-19), was declared a global pandemic by the World Health Organization (WHO). SARS-CoV-2 is an enveloped virus belonging to the genus *Betacoronavirus*, which also includes SARS-CoV-1, the virus responsible for the 2003 SARS outbreak, and Middle East respiratory syndrome coronavirus (MERS-CoV), the virus responsible for the 2012 MERS outbreak. Despite the devastating effects these viruses have had on public health and the economy, distributing vaccines globally has been challenging[1], and the first two therapeutics that substantially reduce mortality of COVID-19 were identified only in late 2021, nearly two years after the disease was discovered[2–4]. There is therefore an urgent need to understand the unique RNA biology of and develop new therapeutics against viruses in this genus.

Coronavirus genomes consist of single-stranded, positive-sense RNA and are the largest among all known RNA viruses (27–32 kb)[5]. Prior to the emergence of SARS-CoV-2, most studies on secondary structures within coronavirus RNA genomes focused on several conserved regions that are essential for viral replication: the 5′ UTR, the 3′ UTR, and the frameshifting stimulation element (FSE)[6,7]. Functional studies have revealed the importance of their secondary structures for viral transcription and replication[6,8–10].

The first roughly two-thirds of every coronavirus genome consists of one open reading frame (ORF1) encoding 16 nonstructural proteins (nsps)[11]. A stop codon near the middle of ORF1 partitions it into an upstream ORF1a and downstream ORF1b. While some ribosomes stop after translating a polyprotein from only ORF1a (nsps 1–11), the FSE causes a fraction of ribosomes to "slip" backward by 1 nt, bypass the stop codon, and translate a polyprotein from the entire ORF1ab (nsps 1–10, 12–16)[6]. Several proteins unique to ORF1ab are central to RNA replication and transcription, including the viral RNA-dependent RNA polymerase (nsp12) and helicase (nsp13)[7,12]. Studies on multiple coronaviruses have shown that an optimal ribosomal frameshifting rate is critical, and small differences in the percentage of frameshifting lead to dramatic differences in genomic RNA production and infectivity. Therefore, the FSE has emerged as a major drug target for small molecules that can influence the rate of ribosome slippage and is under active investigation to be used as a treatment against SARS-CoV-2[13–16].

The structures of coronavirus FSEs have been studied extensively. Short segments of the core FSE from both SARS-CoV-1[7] and SARS-CoV-2[14] fold into a complex structure with a three-stemmed pseudoknot. Small molecules, locked nucleic acids (LNAs), and mutations that are intended to disrupt this structure have been shown to impair viral replication[13–16]. However, despite the importance of the FSE structure, there is to date no direct validation of the relationship between the RNA folding conformation and frameshifting rate in infected cells.

Over the last decade, major advances in methods for RNA chemical probing have enabled genome-wide characterization of RNA structures in living cells. The most commonly used chemical probes are dimethyl sulfate (DMS)[17] and reagents in the SHAPE[18] and icSHAPE[19] families. DMS reacts with the Watson–Crick face of adenine (A) and cytosine (C) bases and probes base-pairing directly, while SHAPE and icSHAPE reagents react with the 2′-OH group of all four nucleotides and measure nucleotide flexibility as a proxy for base pairing[20]. Predictions of RNA structure that use DMS reactivities as folding constraints are of similar or marginally higher accuracies than predictions using SHAPE reactivities, as the specificity of DMS for Watson–Crick base-pairing compensates for the ability of SHAPE to probe all four nucleotides[20].

Three studies[21–23] recently proposed models of the secondary structure of the entire genome of SARS-CoV-2 in human or monkey cells using SHAPE-MaP[18] or icSHAPE[19]. These models are based on the average (ic)SHAPE reactivity at each nucleotide, and cannot provide direct experimental evidence for alternative structures. However, the genomes of RNA viruses form not one structure but an ensemble of many structures whose dynamics regulate critical viral processes, such as splicing in HIV-1[24]. Thus, more work is needed to determine the dynamics of RNA structures within the SARS-CoV-2 genome and their functional roles in the viral life cycle.

In this study, we perform DMS mutational profiling with sequencing (DMS-MaPseq)[25] and DREEM clustering[24] on infected Vero and Huh7 cells to generate experimentally determined, single-nucleotide resolution genome-wide secondary structure ensembles of SARS-CoV-2. Our results reveal major differences with in silico and population-average structure predictions. Importantly, we highlight the physiological structure dynamics of known functional elements, such as the alternative structures at the FSE that determine frameshifting rates in cells. Our work provides experimental data on the structural biology of RNA viruses and will inform efforts on the development of RNA-based diagnostics and therapeutics for SARS-CoV-2.

## Results

**The genome-wide structure of SARS-CoV-2 in cells**. To determine the intracellular genome-wide structure of SARS-CoV-2, we added DMS to infected Vero or Huh7 cells and performed mutational profiling with sequencing (DMS-MapSeq)[25] (Fig. 1a). We chose DMS because it rapidly modifies unpaired adenines (As) and cytosines (Cs) in vivo at their Watson–Crick faces with negligible background effects[25] and has been shown to yield structures of similar or slightly higher accuracies compared to SHAPE[20]. We obtained high genome sequencing coverage (Fig. 1b), and the samples had high signal to noise ratios (Fig. 1c), with adenines and cytosines having a mutation rate ~5-fold higher than the background (guanines and uracils) and over 20-fold higher than all four nucleotides in untreated RNA (0.10%). The results were highly reproducible between independent biological replicates ($r = 0.93$; Fig. 1d), and overall, the DMS reactivities of SARS-CoV-2 were similar in the infected Vero and Huh7 cells ($r = 0.84$, Fig. 1d). We used the DMS-MaPseq data as constraints in RNAstructure[26] to fold the entire SARS-CoV-2 genomic RNA (Supplementary Fig. 1) and assessed the quality of our model using two approaches.

First, the quality of a chemical probing dataset is commonly measured using the area under the receiver operating characteristic curve (AUROC) to evaluate an RNA in the dataset whose structure had been solved previously[23,25]. We determined that AUROC values of roughly 0.95 or higher indicate high-quality probing data by benchmarking two RNAs with known, robust structures for which we had previously collected DMS-MaPseq data:[24] the U4/U6 snRNA (AUROC = 0.98) and the Rev response element (RRE) from HIV-1 (AUROC = 0.95) (Fig. 2a). By contrast, randomly shuffling DMS reactivities 100 times reduced the average AUROC to 0.50 (Fig. 2a). To validate our SARS-CoV-2 datasets directly, we chose stem–loop 5 (SL5) within the 5′ UTR, whose secondary structure has been validated extensively using homology modeling[27], SHAPE-MaP[21,22], icSHAPE[23], RNase and inline probing[28], and NMR[29]. All approaches but NMR yielded exactly the same structure (NMR differed by only two base pairs). Compared to this literature consensus structure, our Huh7 and Vero in-cell DMS-MaPseq datasets yielded AUROC values of 0.99 and 0.98, respectively, (Fig. 2a, b), showing that our in-cell data were of high quality. We note that the other datasets[21–23] yielded lower AUROC values over SL5 (Fig. 2b) and ORF1 (Supplementary Fig. 5). AUROC values for our reactivity data and predicted

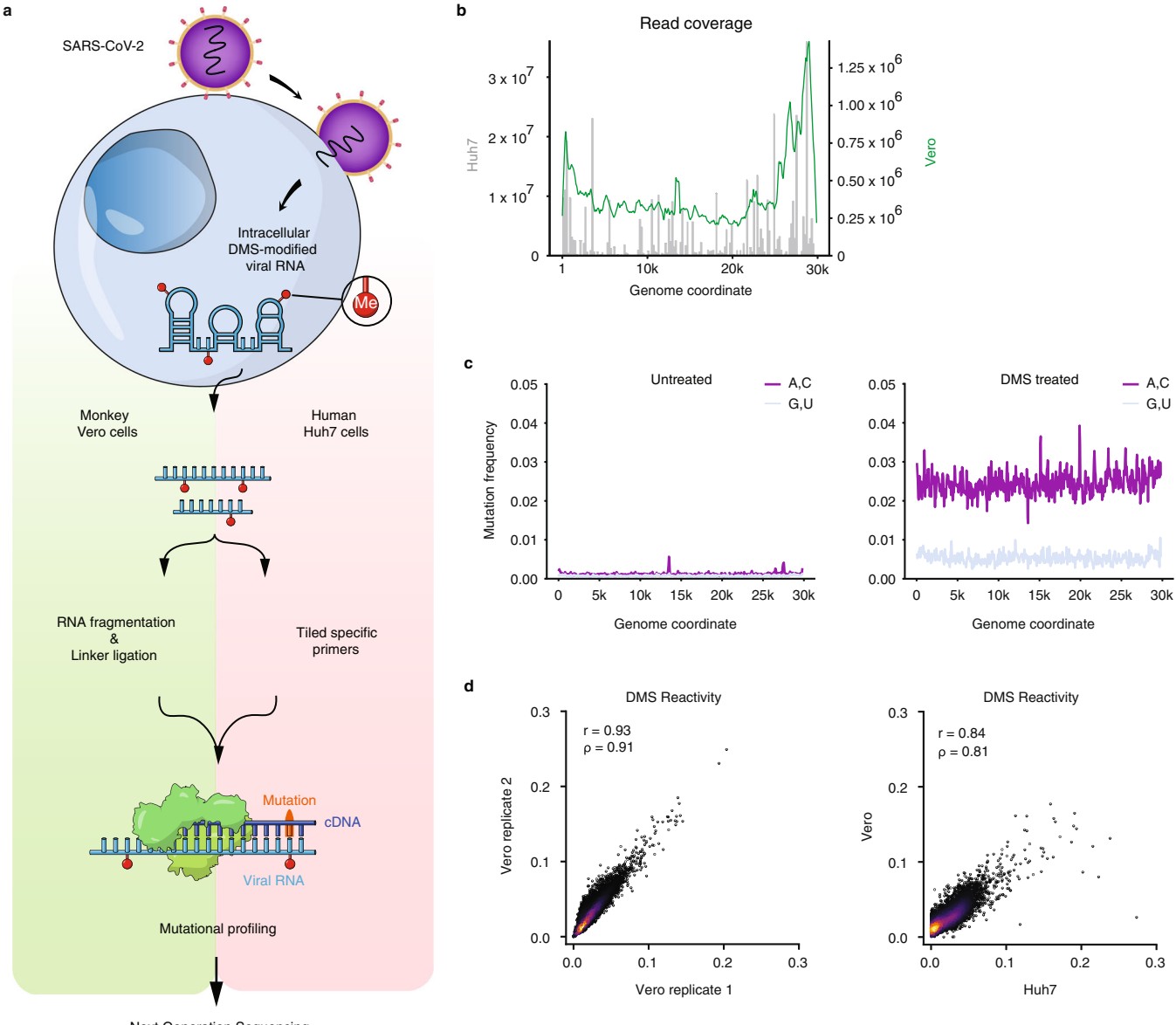

**Fig. 1 Genome-wide probing of SARS-CoV-2 RNA structure in infected Vero and Huh7 cells with DMS-MaPseq. a** Schematic of the experimental protocol for probing severe acute respiratory syndrome coronavirus 2 (SARS-CoV-2) RNA structures in Vero and Huh7 cells using dimethyl sulfate mutational profiling with sequencing (DMS-MaPseq). **b** Read coverage as a function of genome coordinate for Huh7 cells using tiling specific primers (gray bars, left axis) and Vero cells using linker ligation (green curve, right axis); Vero coverage was smoothed by taking the mean over a sliding window of 500 nt. **c** Signal vs. noise plots of mutation frequencies (i.e., among all reads aligning to each genome coordinate, the fraction of reads with a mutation at that coordinate) on adenines (As) and cytosines (Cs) vs. guanines (Gs) and uracils (Us) as a function of genome coordinate for untreated and DMS-treated RNA. A mutation frequency of 0.01 at a given position represents 1% of reads having a mismatch or deletion at that position. Signal and noise were smoothed by taking the mean over 100 nt windows in increments of 50 nt. **d** Comparison of DMS reactivities on As and Cs between biological replicates in Vero cells (left) and between the averaged of Vero replicates and Huh7 cells (right). Pearson ($r$) and Spearman ($\rho$) correlation coefficients are shown. For each sample, the top 0.05% of mutational fractions (values over 0.27 for Vero and 0.38 for Huh7) were considered outliers and excluded from the plot and calculation of correlation coefficients. Source data are provided as a Source Data file.

structures genome-wide indicated that the Huh7 dataset was high-quality (AUROC = 0.95) and the Vero dataset moderately high quality (AUROC = 0.89) (Fig. 2a).

Second, we found that our models of the 5′ untranslated region (UTR) (Figs. 2c) and 3′ UTR (Supplementary Fig. 7a) agreed well with previous studies, showing that we could accurately identify known secondary structures. The secondary structures of the 5′ UTR are conserved in multiple coronaviruses and have been characterized extensively[6,21,22,27,28,30]. In agreement with previous studies, we found five stem–loops (SL1–5) within the 5′

UTR (nucleotides 1–265). These structures perform essential functions in viral replication (SL1[8] and SL2[9]), subgenomic RNA production (SL3[6] and SL4[31]), and escape of nsp1-mediated translational suppression (SL1[32]). SL5 contains the start codon of ORF1 and branches into three additional stems (SL5A, SL5B, and SL5C), which our model recapitulates perfectly with respect to previous studies[21,28]. In agreement with another in-cell structure model[22], we did not find evidence for a short stem–loop (SL4.5) proposed in several other studies[21,27,28]. Additional structures exist immediately downstream of the 5′ UTR. We found three

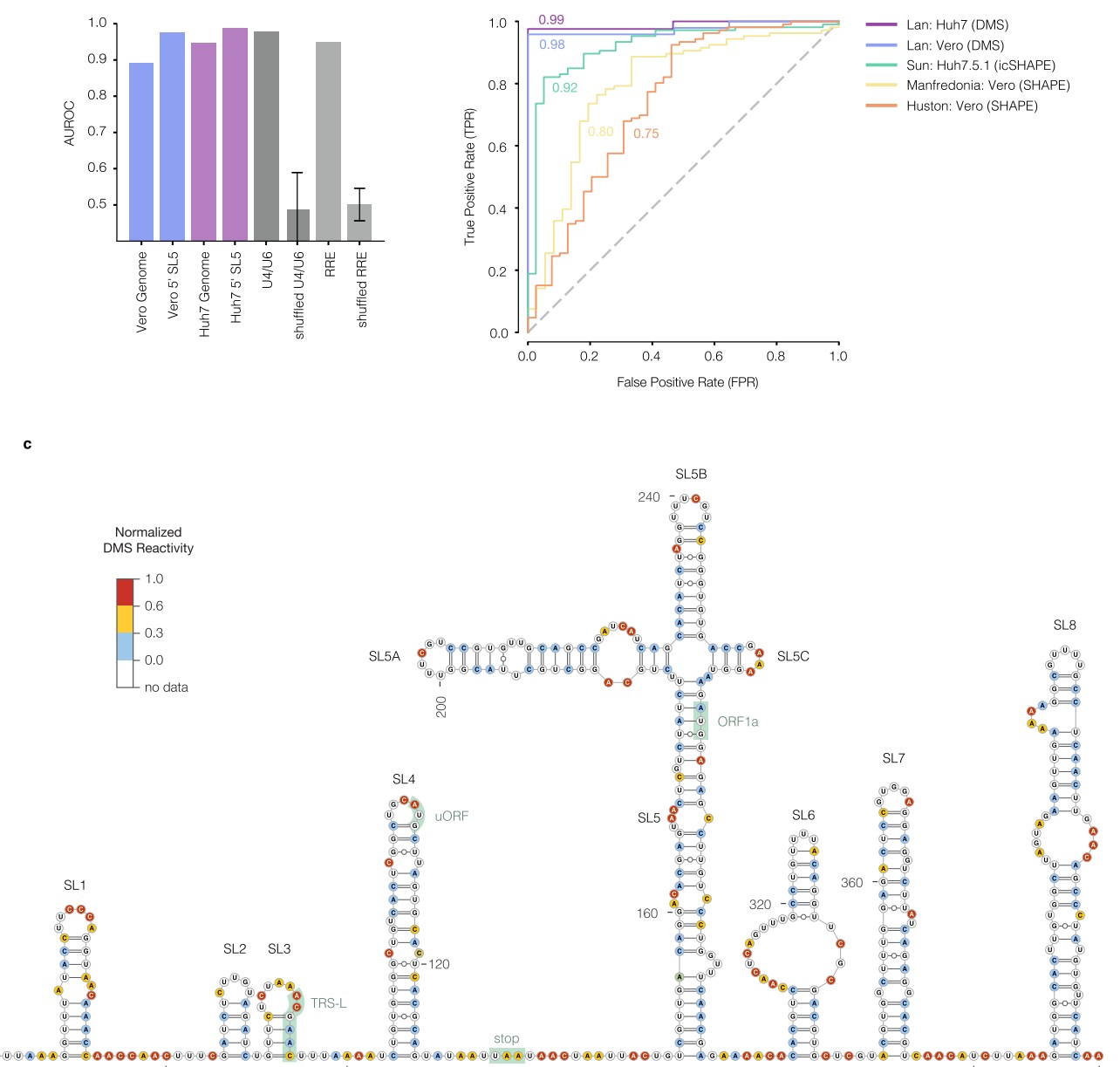

**Fig. 2 Quality assessment of the SARS-CoV-2 secondary structure model. a** Agreement between DMS reactivities and predicted structures for the Vero and Huh7 genomes, and the consensus structure of 5' untranslated region (UTR) stem–loop 5 (SL5; coordinates 150–294), measured as the area under the receiver operating characteristic curve (AUROC). AUROC values between DMS-MaPseq data and well-established structures are also shown for two positive control RNAs: U4/U6 snRNA and HIV-1 Rev Response Element (RRE). As negative controls, $n = 100$ shuffled datasets were generated by randomly permuting the DMS reactivities and recomputing the AUROC. Data are presented as means ± SD. **b** Receiver operating characteristic (ROC) curves comparing the literature consensus structure of SARS-CoV-2 SL5 (coordinates 150–294) with DMS/SHAPE reactivities from our datasets and those from other authors. Each AUROC value is shown next to its ROC curve. For each dataset, the first author, cell type, and chemical probe are indicated. **c** Model of the first 480 nt of the SARS-CoV-2 genome (including the 5' UTR, coordinates 1–265) based on DMS reactivities from Vero cells. Nucleotides are colored by normalized DMS reactivities. Highlighted features include stem–loops (SL) 1–8, the leader TRS (TRS-L), the start codons of the upstream ORF (uORF) and ORF1a, and the stop codon of uORF. Source data are provided as a Source Data file.

stem–loops (SL6–8) in this region, in nearly perfect agreement with two previous in-cell studies[21,22]. In further support of the accuracy of our model, the AUROC was 0.93 across SL1–5, indicating that our model of this region agrees very well with our chemical probing data (Supplementary Fig. 2c).

**Genome structures that are well supported by multiple lines of evidence**. To identify structures within the genome that are well

supported by multiple lines of evidence, we compared our population average model with those from published studies. RNA structures are commonly compared using sensitivity and positive predictive value (PPV)[33], whose mean is the Fowlkes-Mallows index (FMI)[34]. We introduced a modified FMI (mFMI) that also accounts for bases that are unpaired in both structures (see Methods). We determined regions of local similarity by computing the mFMI across the genome and found multiple structures that are supported across published studies

(Supplementary Fig. 4). To facilitate future studies investigating the binding of LNA probes to the genome, we also determined the locations of all stretches of at least 14 consecutive unpaired bases (Supplementary Fig. 8a).

As the transcription-regulating sequences (TRSs) are necessary for the synthesis of sgRNAs, we analyzed our structural models of the leader TRS (TRS-L) and the nine body TRSs (TRS-B). The leader TRS (TRS-L) is the central component of the 5′ UTR involved in discontinuous transcription[11] and is predicted to lie in a short stem–loop (SL3), consistent with the previous results[21–23,27]. Of the nine body TRSs, we find that seven (all but the TRSs of ORF7a and ORF7b) lie within a stem–loop (Supplementary Fig. 8b).

To further support our genome-wide structure model, we analyzed covariation between paired bases, which has been used previously to support the existence of RNA structures[22,35–37]. Most methods for analyzing covariation in RNA were developed for non-coding RNAs, although covariation is also present at the amino acid level[38], which could confound the analysis of RNA structure. With this caveat in mind, we divided our genome-wide model into 353 structural elements encompassing 75% of the genome, built a covariance model for each element using Infernal[37], and identified base pairs supported by covariation using R-scape[36] with a database of 301,535 non-redundant, full-length coronavirus genomes. We detected 95 base pairs supported by covariation at an E-value threshold of 0.05 (Supplementary Data 1). At least one covarying pair was present in 63 structural elements, and 18 were supported by at least two pairs, providing further support for our genome-wide structural model (Supplementary Data 2). Among the elements with the most covarying pairs were SL8 downstream of the 5′ UTR (two pairs), the stem containing s2m in the 3′ UTR (four pairs), and a short, unannotated hairpin near the 5′ end of the N gene (five pairs).

**At least half of the SARS-CoV-2 genome forms alternative structures.** We previously discovered that for another ssRNA virus, HIV-1, over 90% of the genome forms ensembles of alternative structures rather than a single structure[24]. The formation of alternative RNA structures has important functional consequences: for example, in HIV-1, they regulate alternative splicing. However, all previous studies that chemically probed the entire SARS-CoV-2 genome in cells[21–23] used only the average reactivity of each base to fold their structural models, and thus could not detect subpopulations of RNAs with different structures. Although these studies used Shannon entropy to estimate structural heterogeneity in a series of short sliding windows, this metric is still based on the average SHAPE reactivities per base and does not identify subpopulations of alternative structures directly from single-molecule data.

We detected alternative structures in SARS-CoV-2 by applying the DREEM algorithm[24] to our in-cell DMS-MaPseq datasets. Briefly, DREEM clusters the sequencing reads based on which bases are DMS modified together on the same read and identifies sub-populations of molecules with distinct patterns of DMS modifications.

Our data from both Vero and Huh7 cells reveal that the majority of the SARS-CoV-2 genome forms alternative structures. We hypothesized that if a region forms at least two very distinct alternative structures, the local agreement between the DMS reactivities and the population average model (i.e., the AUROC) would tend to be low, and vice versa. Consistent with our hypothesis, AUROC and ΔDMS correlated negatively ($r = -0.32$, $P < 10^{-16}$, two-tailed beta distribution test), albeit weakly, indicating that large differences between alternative structures are associated with lower agreement between the population

average structure and the DMS reactivities (Supplementary Fig. 9a). This application of AUROC—evaluating the quality of predicted structures based on reactivity data—inverts its traditional use in evaluating the quality of reactivity data based on ground truth structures. To further justify this new application, we generated decoy structures of the U4/U6 snRNA and RRE (see Methods) and computed the AUROC and similarity to the ground truth structures (Supplementary Fig. 9b). Decoys highly similar to the true structure tended to have high AUROC, but decoys much different from the true structure had a wider range of AUROC. Therefore, a low AUROC indicates a large deviation from the true structure, but a high AUROC does not necessitate that a predicted structure is correct. Thus we use AUROC as a measure to identify incorrect structures that are not well supported by the underlying data.

We, therefore, reasoned that the genomic regions that would benefit most from representation as an ensemble of structures (rather than as a single population average) would be those with high ΔDMS (i.e., large differences between clusters) and low AUROC (i.e., a poor agreement between population average structure and reactivities). From our Vero model, we identified all coordinates at which the AUROC and ΔDMS were, respectively, below and above their genome-wide medians. Denoising these coordinates with a low-pass filter (see Methods) yielded 69 regions best represented as structural ensembles, covering 24% of the genome (Fig. 3). For Huh7 cells, we used an orthogonal approach involving three very stringent filters (see Methods) and identified 71 regions that formed at least two alternative structures, covering 37% of the genome (Fig. 3, Supplementary Data 3, Supplementary Data 4). Combined, the regions forming alternative structures in either model covered 50% of the genome, highlighting the prevalence of alternative structures. ORF1ab was slightly enriched for alternative structures (55% covered by either model) relative to the downstream ORFs (38% covered).

Interestingly, we found that the FSE, which is critical for regulating the translation of ORF1b, meets the above criteria for alternative structures in both Vero and Huh7 cells (Fig. 3, red highlight). Although other studies have suggested that the FSE forms multiple structures, they have either inferred them indirectly using suboptimal folding based on population average reactivities[21] or measured them using short segments of the FSE and/or in vitro, outside of the context of genomic RNA and cellular factors[39]. We find that the FSE indeed forms at least two distinct structures, consistent in Vero and Huh7 cells, and characterize them in detail below.

**Uncovering an unexpected structure at the FSE.** The FSE causes the ribosome to slip and shift register by −1 nt in order to bypass a stop codon and translate ORF1b, which encodes five non-structural proteins (nsps) including nsp12, an RNA-dependent RNA polymerase (RdRP)[40]. Controlling the rate of frameshifting is thought to be essential for viral viability[41], and thus many studies have used small molecules or antisense oligonucleotides to target the canonical three-stemmed pseudoknot structure of the FSE to attenuate viral fitness[13–16].

This pseudoknot structure, which is thought to promote frameshifting, was identified by analyzing short constructs with chemical probing, nuclease mapping, nuclear magnetic resonance (NMR), and cryo-electron microscopy (cryo-EM)[7,14,42,43]. We also in vitro-transcribed, refolded, and DMS-probed an 88 nt segment of the SARS-CoV-2 FSE similar to those used in the previous studies[7,14]. Our in vitro data-driven model for the predominant structure of this RNA (Fig. 4a, top) agrees well with the previous models in that it finds all three canonical stems, including the pseudoknot. However, we were particularly

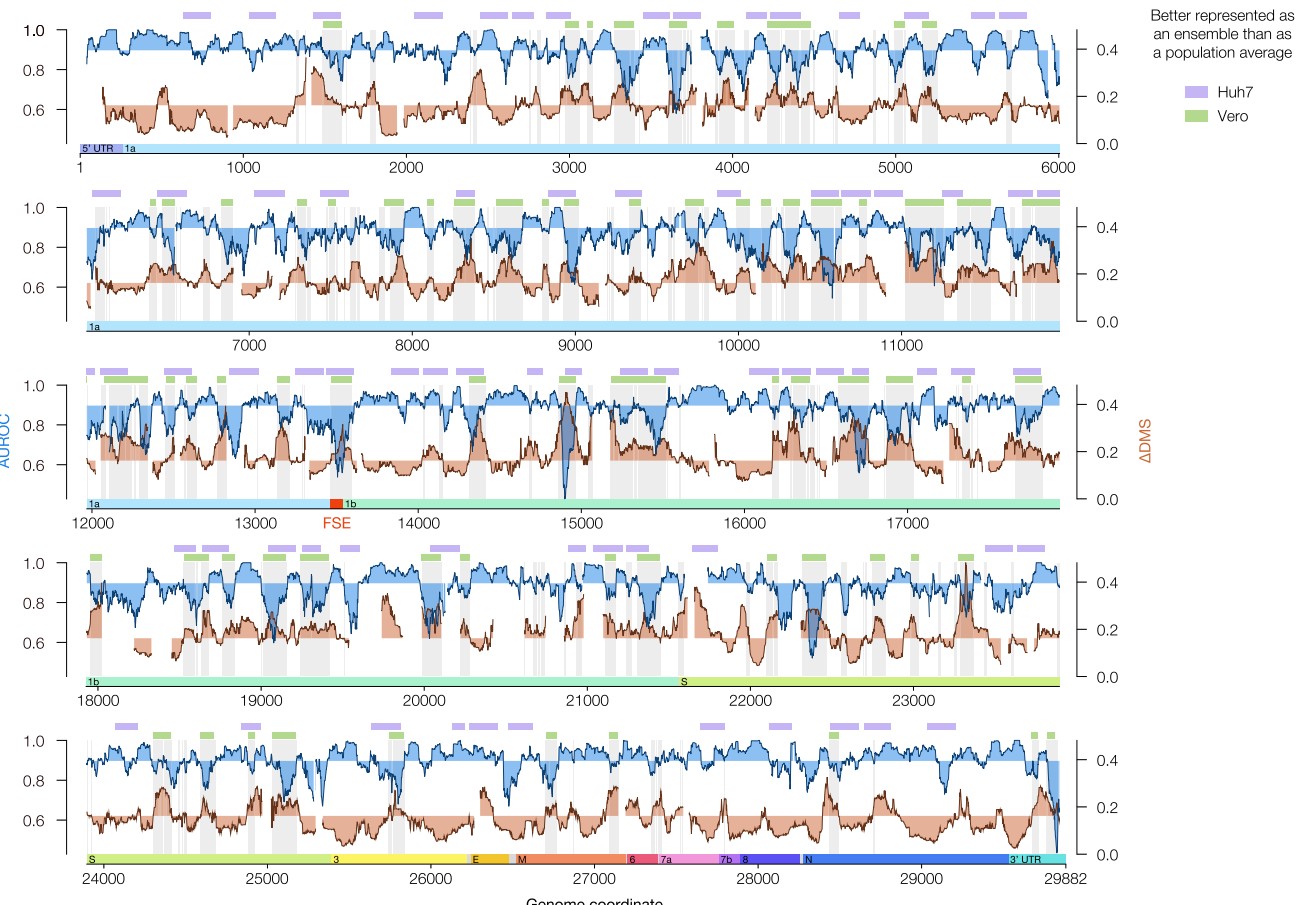

**Fig. 3 Alternative RNA structures form across the SARS-CoV-2 genome.** Agreement between DMS reactivities and predicted secondary structures (AUROC, blue) and the difference in DMS reactivity between clusters 1 and 2 (ΔDMS, orange) for the genome-wide model in Vero. Both quantities were calculated over sliding windows of 80 nt in 1 nt increments; *x* values represent the centers of the windows. Windows with <10 paired or <10 unpaired bases were excluded from the calculation of AUROC; windows with <10 bases that clustered into at least two structures were excluded from the calculation of ΔDMS. For AUROC and ΔDMS, the area between the local value and the genome-wide median is shaded. For the Vero model, all coordinates best described by structure ensembles (AUROC below median, ΔDMS above median) are shaded in light gray. The green bars represent a denoised version of these coordinates (see Methods). For the Huh7 model, regions meeting criteria for alternative structures (see Methods) are labeled with lavender bars. The locations of the untranslated regions (UTRs) and open reading frames (ORFs) of SARS-CoV-2 are indicated below the AUROC and ΔDMS data. The frameshifting stimulation element (FSE, coordinates 13,462–13,546) is highlighted in red. Source data are provided as a Source Data file.

interested in this structure not just in vitro but in the context of the full genome in infected cells.

To closely examine the FSE structure in cells, we used the target-specific DMS-MaPseq protocol[25]. We designed primers targeting 283 nt surrounding the FSE and amplified this region from SARS-CoV-2 infected Vero and Huh7 cells treated with DMS. Our analysis revealed a strikingly different structure than the prevailing model[7,27] (Fig. 4a, bottom). In our in-cell model, the expected pseudoknot does not form downstream of the slippery sequence. Instead, a sequence that partially overlaps stem 1 of the canonical pseudoknot (Fig. 4a, bottom, pink) pairs with a 10 nt perfectly complementary sequence upstream of the slippery site (Fig. 4a, bottom, blue). We call this pairing Alternative Stem 1 (AS1).

Interestingly, in silico predictions of the RNA structure of the SARS-CoV-2 genome using RNAz[27] and ScanFold[44] also support our in-cell model of Alternative Stem 1. Additionally, both studies that have probed the structure of the SARS-CoV-2 FSE in infected cells[21,22] found that their chemical reactivity data disagreed with the three-stemmed pseudoknot. Thus, a variety of computational predictions and chemical probing experiments all favor Alternative Stem 1 over the three-stemmed pseudoknot

as the predominant structure of the FSE in the context of the full viral genome.

**AS1 pairing sequence is conserved across SARS-related coronaviruses.** To determine if other coronaviruses may have a similar alternative structure of the FSE, we searched for the sequence that pairs with canonical stem 1 in a set of curated coronaviruses[45]. This set contains 53 isolates of SARS-CoV-2, 12 other SARS-related coronaviruses, and 2 MERS-related coronaviruses. All 10 nt of the 5′ side of AS1 were perfectly conserved in all 12 of the SARS-related viruses, including SARS-CoV-1 and six viruses isolated from bats (Fig. 4b, purple). However, the 10 nt complement was not present in either MERS-related viruses. Among the 20 betacoronaviruses with complete genomes in RefSeq[46], the 10 nt complement was present in only the three SARS-related viruses (SARS-CoV-1, SARS-CoV-2, and BtCoV BM48-31). These results suggest that AS1 is unique to SARS-related coronaviruses.

**The FSE structure is dependent on the sequence context.** The major differences we observed in the structure of the FSE in cells

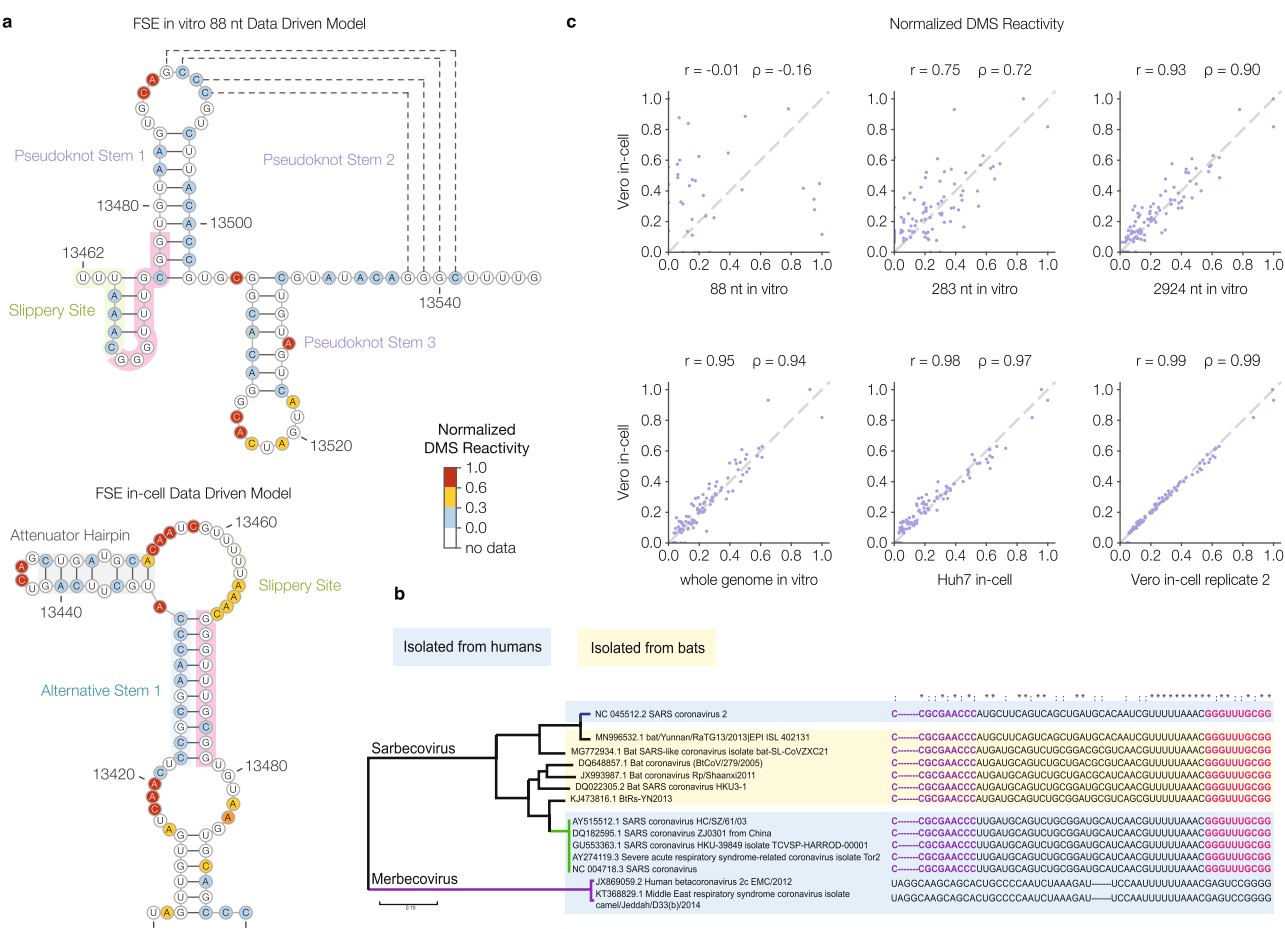

**Fig. 4 The frameshifting stimulation element (FSE) adopts an unexpected structure in cells. a** Predicted structures of the FSE derived from DMS-MaPseq on in vitro-transcribed 88 nt RNA (top) and infected Vero cells (bottom). For the 88 nt RNA, reads were clustered into $K = 3$ clusters; in the cluster with the largest fraction of reads (60%), the given pseudoknot was among the three minimum-energy structures. Nucleotides are colored by normalized DMS reactivities (see Methods). The 5′ and 3′ sides of the alternative stem 1 (AS1) are highlighted in blue and pink, respectively (bottom), and the sequence that forms the 3′ side of AS1 is also highlighted in pink in the top structure. **b** Sequence conservation of FSE alternative stem 1 pairing. The 5′ and 3′ sequences of alternative stem 1 are highlighted in purple and pink, respectively. Symbols above the sequences indicate perfect conservation among all viruses in the alignment (*) or perfect conservation among only the sarbecoviruses (:). Source data are provided as a Source Data file. **c** Scatterplots of DMS reactivities over the FSE, comparing infected Vero cells and other contexts: 88 nt, 283 nt, or 2924 nt RNA fragments containing the FSE folded in vitro; whole genomic RNA extracted from virions and refolded in vitro; infected Huh7 cells; and a replicate of infected Vero cells. In each sample, DMS reactivities have been normalized by dividing by the maximum reactivity. For each comparison, the Pearson ($r$) and Spearman ($\rho$) correlation coefficients are given.

vs. in vitro could either be due to 1) length of the in vitro refolded viral RNA or 2) factors in the cellular environment that is absent in vitro. To distinguish between these two possibilities, we refolded the FSE in vitro in the context of longer native sequences.

We found that as we increased the length of the in vitro refolded construct by including more of its native sequence, from 88 nt to the whole genome (~30 kb), the DMS reactivity patterns became progressively more similar to the pattern we observed in cells (Fig. 4c). Indeed, in the context of the full ~30 kb genomic RNA, the DMS reactivities of the in vitro folded FSE are nearly identical to those during SARS-CoV-2 infection in Vero and Huh7 cells ($r = 0.95$). These results show that the FSE folds correctly in the absence of protein factors. Importantly, the upstream and downstream sequence is necessary for folding the FSE, suggesting the presence of long-range RNA:RNA interactions (Fig. 4c).

**The FSE forms alternative structures in cells**. We further analyzed the intracellular folding of the FSE using DREEM. We found

two distinct patterns of DMS reactivities (Fig. 5a), showing that the RNA folds into at least two distinct conformations at this region. These major conformations were identical in both Huh7 and Vero cells (Fig. 5b). Surprisingly, we found that Cluster 2 (45% abundance) corresponds to a very long-range RNA:RNA interaction that spans ~1.2 kb of sequence (Fig. 5c). Many of the base pairs of this interaction are also supported by psoralen crosslinking[47].

**Frameshifting rate is determined by FSE sequence context and structure**. To directly measure how the FSE structure ensemble impacts frameshifting rate in cells, we constructed dual-luciferase frameshift reporter constructs[48]. We used either a "short" FSE of only a 92 nt region that folds into the canonical three-stemmed pseudoknot or a "long" 2924 nt sequence containing the FSE, which folds nearly identically as the full-length genome in infected cells (Fig. 4c). A dual-luciferase reporter is a well-established tool for measuring frameshifting rate, where the stop codon of a firefly luciferase (Fluc) coding sequence is replaced with an FSE which allows a Renilla luciferase (Rluc) coding

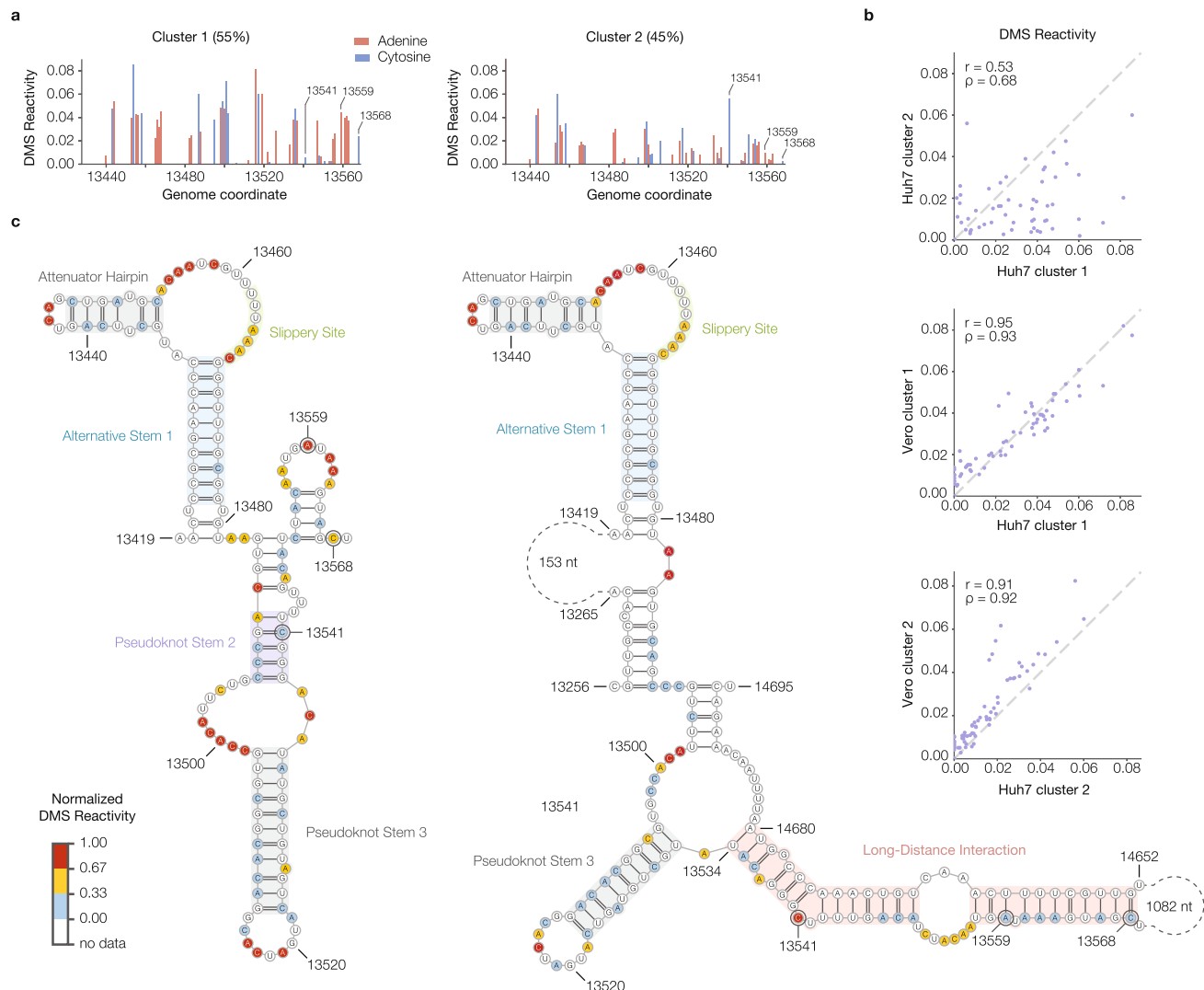

**Fig. 5 Alternative conformations of the frameshifting stimulation element (FSE) derived from in-cell DMS-MaPseq data include a long-distance interaction.** **a** DMS reactivity profiles for both clusters from the Huh7 genome-wide RT-PCR data in the vicinity of the FSE (nucleotides 13,434–13,568). The abundance of each cluster is given beside its name. Each bar representing an adenine or cytosine is colored in red or blue, respectively. Three of the nucleotides whose reactivities differ substantially between clusters are labeled. **b** Scatterplots of DMS reactivities over the FSE, comparing the two clusters from Huh7 (top) and each Huh7 cluster with the corresponding cluster from Vero cells (middle and bottom). For each comparison, the Pearson ($r$) and Spearman ($\rho$) correlation coefficients are given. **c** Predicted structures of Huh7 clusters 1 and 2 based on DMS reactivities. In each structure, selected features are highlighted, including alternative stem 1 (in both clusters), a long-distance interaction (in cluster 2), and features that are also present in the canonical pseudoknot. The three nucleotides labeled in (**a**) are also labeled in the structure models. Nucleotides are colored by normalized DMS reactivities. Source data are provided as a Source Data file.

sequence in the −1 frame behind the FSE to report on frameshifting rate (Fig. 6a). In addition, we in vitro transcribed and transfected the reporter mRNA into cells to avoid cryptic transcription start sites or unintended splicing events of the DNA reporter that could impact Fluc and Rluc luminescence. In-cell DMS probing was performed to ensure that the addition of luciferase does not change the structure of the FSE (Supplementary Fig. 11c). We calculated the frameshifting rate as the relative Rluc to Fluc ratio after normalization against amino acid-matched negative and positive controls.

Previous studies using similar constructs have focused on just the short FSE and found that it promotes 20–30% frameshifting[13,16]. Strikingly, we found that the long FSE frameshifted at ~42% while the short FSE frameshifted at only ~17% (Fig. 6b). Our results on the long FSE are in agreement with in vivo ribosome profiling measurements of SARS-CoV-2 infected cells[49] (Fig. 6c), indicating that the previously predicted

structure of the canonical 92 nt FSE does not recapitulate the mechanism of ribosomal frameshifting on the full-length virus during infection. Although additional studies are needed to understand the precise nature of the interactions between sequences further up and downstream in ORF1a and ORF1b that impact both the FSE structure ensemble and frameshifting rate (Fig. 6c), our results underscore the importance of studying RNA structure ensembles in cells and in its full-length context.

## Discussion

Here, we present insights into the secondary structure ensembles of the entire SARS-CoV-2 RNA genome in infected cells based on chemical probing with DMS-MaPseq. Previous work on the RNA structures of SARS-CoV-2 has provided only population-average models, which assume that the RNA folds into one conformation. In addition to our population-average model, we used the

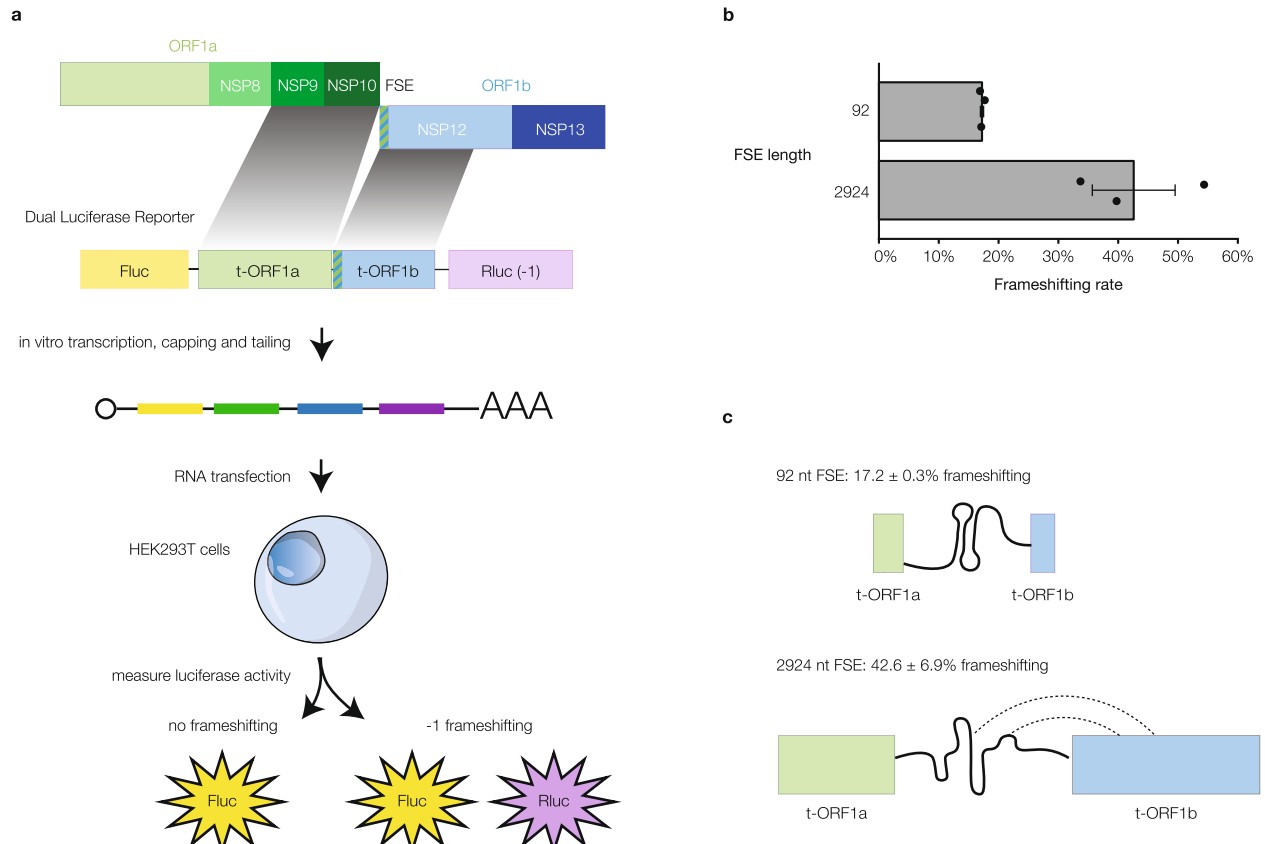

**Fig. 6 The long frameshifting stimulation element (FSE) has a dramatically higher frameshifting rate than the minimal FSE. a** Schematic of the 2924 nt dual-luciferase construct containing the FSE. The construct consists of truncated parts of segments a and b of open reading frame 1 (t-ORF1ab) encoding non-structural proteins (nsps) nsp9, nsp10, and most of nsp12 inserted between firefly luciferase (Fluc) in reading frame 0 and Renilla luciferase (Rluc) in reading frame −1. With −1 frameshifting, both Fluc and Rluc are expressed; without, only Fluc is expressed. **b** Rate of −1 ribosomal frameshifting calculated as Rluc/Fluc normalized against amino-acid matched positive and negative controls for both 92 nt and 2924 nt inserts for $n = 3$ biologically independent experiments. Data are presented as mean values ± SEM. $P = 0.053$ for difference in means, two-sided Welch's $t$-test. **c** Schematic of the RNA structure ensemble speculated to lead to a higher frameshifting rate. Source data are provided as a Source Data file.

clustering algorithm DREEM[24] to detect and quantify alternative structures genome-wide, revealing previously undescribed conformations at critical positions including the FSE.

Our DMS-MaPseq/DREEM framework gives data with the highest reproducibility and agreement with predicted structures, compared to all other chemical probing work on the SARS-CoV-2 genome to date[21–23]. Importantly, our approach is unique in detecting RNA structural heterogeneity directly from the data itself, without prior assumptions about RNA folding. We previously validated DMS-MaPseq/DREEM on gold standard structures[24], and now we generate a secondary structure model for the entire SARS-CoV-2 genomic RNA, highlighting regions that fold into single structures and heterogeneous ensembles. Structural heterogeneity in and downstream of the S gene (coordinates 21,556 – 29,882) may result either from each species of genomic and subgenomic RNA forming a distinct structure, or all species forming similar ensembles of alternative structures. Although our data cannot distinguish between these possibilities, one study suggests that genomic and subgenomic RNAs form similar structures, except for two regions (27,750–850 and 29,200–300)[50]. Within both of these regions (Fig. 3), we discovered structural heterogeneity that may correspond to sgRNA-specific structures, while the other regions of heterogeneity are more likely to be ensembles that are similar among multiple sgRNAs.

Our in-cell data reveal alternative conformations of the FSE within its genomic sequence context distinct from the canonical pseudoknot seen when considering only the minimal 88 nt FSE. We show that in vitro RNA-refolding of the full-length 30 kb genome can recapitulate the structure ensemble formed at the FSE in cells. Importantly, we show that the longer sequence is critical to achieving the frameshifting rate observed in cells during viral infection. When used in dual-luciferase reporters, the longer sequence (3 kb) frameshifts at a much higher rate than the minimal FSE (~42% compared to ~17% of the minimal sequence). These results underscore a functional role for long-range RNA interactions[47] and explain data from recent ribosome profiling studies showing that ribosomes frameshift at >50% in infected cells[49,51].

Our in-cell data-derived model of SARS-CoV-2 presents major RNA structures and sites of RNA structure heterogeneity across the entire genome and provides the foundation for further studies. Importantly, our work reveals that drugs such as small molecules or antisense oligos intended to abolish SARS-CoV-2 frameshifting should be designed and tested against the correct structure ensemble that forms in cells. Further work to better understand the functional significance of other structured elements across the SARS-CoV-2 genome will enable the design of more targeted therapeutics.

## Methods

**Cell culture and SARS-CoV-2 infection**. Monkey Vero cells (ATCC CCL-81) cultured in DMEM (Gibco) supplemented with 10% fetal bovine serum (FBS) (Gibco) and 1% 260 Penicillin/Streptomycin were plated onto 100 mm dishes and

infected at an MOI of 0.01 with 2019-nCoV/USA-WA1/2020 (Passage 6). Infected cells were incubated at 37 °C, 5% CO$_2$ for 48 h before DMS treatment.

Human Huh7 cells were cultured in Dulbecco's Modified Eagle 259 Medium (DMEM) supplemented with 10% FBS and 1% 260 penicillin/streptomycin. Cells were maintained at 37 °C in 5% CO$_2$. Cells were infected with 2019-nCoV/USA-WA1/2020 (Passage 2) at a MOI of 0.05 for 24 h then treated with DMS.

The 2019n-CoV/USA_WA1/2020 isolate of SARS-CoV-2 was obtained from the US Centers for Disease Control and Prevention. Infectious stocks were produced by inoculating Vero E6 cells and collecting the cell culture media upon observation of cytopathic effect; debris was removed by centrifugation and passaged through a 0.22 μm filter.

**DMS modification of SARS-CoV-2 RNA in infected Vero/Huh7 cells.** For Vero cells, 200 μl DMS (or 2% v/v) was added dropwise to the cells and incubated for 4 min at 37 °C. For Huh7 cells, 250 μl DMS (or 2.5% v/v) was added dropwise to the cells and incubated for 3 min at 37 °C. DMS treated cells were then neutralized with 15 ml 30% β-mercaptoethanol in PBS (ThermoFisher Scientific). The cells were centrifuged at 1000*g* for 5 min at 4 °C and then washed twice with 15 ml 30% β-mercaptoethanol in PBS and once with just PBS. Cell pellet was resuspended in 1 ml TRIzol (ThermoFisher Scientific).

**Total RNA extraction and rRNA subtraction.** RNA was extracted following the manufacturer's specifications with the aqueous phase transferred to a new tube and equal volume 100% EtOH added. Total RNA was then purified using RNA Clean and Concentrator −25 kit (Zymo) following the manufacturer's specifications.

Fifteen micrograms of total RNA per reaction were used as the input for rRNA subtraction. First, 1 μl rRNA subtraction mix (15 μg/μl) and 2 μl 5× hybridization buffer (end concentration: 200 mM NaCl, 100 mM Tris-HCl, pH 7.4) were added to each reaction, and the final volume was then adjusted with water to 10 μl. The samples were denatured at 95 °C for 2 min and then the temperature was reduced by 0.1 °C/s until the reaction was at 45 °C. Next, 10 μl RNase H buffer and 2 μl hybridase thermostable RNase H (Lucigen) preheated to 45 ° was added. The samples were incubated at 45 °C for 30 min. The RNA was cleaned with RNA Clean and Concentrator −5, following the manufacturer's instructions, and eluted in 45 μl water. Then, 5 μl Turbo DNase buffer and 3 μl Turbo DNase (ThermoFisher Scientific) were added to each reaction and incubated for 30 min at 37 °C. The RNA was purified with RNA Clean and Concentrator −5 (Zymo) following the manufacturer's instructions.

**Ligation-based DMS-MaPseq library generation.** After rRNA subtraction (described above), extracted DMS-modified RNA was fragmented using the RNA Fragmentation kit (ThermoFisher Scientific). Totally, 1.5 μg of rRNA subtracted total RNA was fragmented at 70 °C for 2.5 min. The fragmented RNA was mixed with an equal volume 2× Novex TBE–urea sample buffer (ThermoFisher Scientific) and run on a 10% TBE–urea gel (ThermoFisher Scientific) at 200 V for 1 h 15 min for size selection of RNA that is ~150 nt. To dephosphorylate and repair the ends of randomly fragmented RNA, 2 μl 10× CutSmart buffer (New England Biolabs), 10 μl shrimp alkaline phosphatase (New England Biolabs), 2 μl RNaseOUT (ThermoFisher Scientific), and water were added to a final volume of 20 μl and 37 °C for 1 h. Next, 4 μl 50% PEG-800 (New England Biolabs), 4 μl 10× T4 RNA ligase buffer (New England Biolabs), 4 μl T4 RNA ligase, truncated KQ (England Biolabs), and 2 μl linker were added to the reaction and incubated for 18 h at 22 °C. The RNA was purified with RNA Clean and Concentrator −5 following the manufacturer's instructions for recovery of all fragments and eluted in 10 μl water. The excess linker was degraded by adding 2 μl 10× RecJ buffer (Lucigen), 1 μl RecJ exonuclease (Lucigen), 1 μl 5′ deadenylase (New England Biolabs) and 1 μl RNaseOUT, then incubating for 1 h at 30 °C. The RNA was purified with RNA Clean and Concentrator −5, following the manufacturer's instructions, and eluted in 11 μl water.

For reverse transcription, 1.5 μg of rRNA subtracted total RNA or 10 μg of in vitro-transcribed RNA was added to 4 μl 5× first strand buffer (ThermoFisher Scientific), 1 μl 10 μM reverse primer, 1 μl dNTP, 1 μl 0.1 M DTT, 1 μl RNaseOUT and 1 μl TGIRT-III (Ingex). The reverse-transcription reaction was incubated at 60 °C for 1.5 h. 1 μl 4 M NaOH was then added and incubated at 95 °C for 3 min to degrade the RNA. The reverse-transcription product was mixed with an equal volume 2× Novex TBE-urea sample buffer (ThermoFisher Scientific) and run on a 10% TBE–urea gel (ThermoFisher Scientific) at 200 V for 1 h 15 min for size selection of cDNA that is ~250 nt. The size-selected and purified cDNA was circularized using CircLigase ssDNA ligase kit (Lucigen) following manufacturer's instructions. 2 μl of the circularized product was used for amplification and addition of custom Illumina adaptors using Phusion High-Fidelity DNA Polymerase (NEB) for a maximum of 16 cycles (Supplementary Data 5). The PCR product was run on an 8% TBE gel at 180 V for 1 h and size-selected for products ~300 nt. The product was then sequenced with NovaSeq6000 system (paired-end run,150 cycles).

**Target sequence-based DMS-MaPseq library generation.** After rRNA subtraction and purification (described above), primers with Tm higher than 60 °C were used directly to reverse transcribe the extracted DMS-modified RNA as described above without fragmentation and linker ligation. For the whole genome rt-PCR library, primers were designed to cover the entire 30 kb with ~50 nt overlap (Supplementary Data 5). The cDNA was then purified with Oligo Clean and Concentrator −5 (Zymo) following the manufacturer's instructions. Two microlitres of cDNA were amplified using Advantage HF 2 DNA polymerase (Takara) for 30 cycles according to the manufacturer's instructions and purified by DNA Clean and Concentrator −5 (Zymo) following the manufacturer's instructions. RNA-seq library for 150 bp insert size was constructed following the manufacturer's instruction (NEBNext Ultra™ II DNA Library Prep Kit) and sequenced on a Nextseq system (paired-end run, 150 cycles).

**In vitro transcribed FSE and DMS modification.** gBlocks were obtained from IDT for the SARS-CoV-2 88 and 283 nt FSE which corresponds to nucleotides 13,459–13,546 and nucleotides 13,342–13,624 based on 2019-nCoV/USA-WA1/2020. The 2924 nt sequence (nucleotides 12,686–15,609) was reverse-transcribed from SARS-CoV-2 and cloned into pmirGLO. The regions of interest were amplified by PCR with a forward primer that contained the T7 promoter sequence (primers 306 + 307, 308 + 309, 310 + 311, respectively; see Supplementary Data 5). The PCR product was used for T7 Megascript in vitro transcription (ThermoFisher Scientific) according to the manufacturer's instructions with a 16 h incubation time at 37 °C. Subsequently, 1 μl Turbo DNase I (ThermoFisher Scientific) was added to the reaction and incubated at 37 °C for 15 min. The RNA was purified using RNA Clean and Concentrator −5 kit (Zymo).

To DMS modify, 10 μg of RNA in 10 μl H$_2$O was denatured at 95 °C for 1 min then placed on ice. On the basis of the DMS concentration used in the next step, 300 mM sodium cacodylate buffer (Electron Microscopy Sciences) with 6 mM MgCl$_2$+ (refolding buffer) was added so that the final volume was 100 μl. (e.g., for 2.5% final DMS concentration: add 87.5 μl refolding buffer and 2.5 μl DMS) Then, 2.5 μl was added and incubated at 37 °C for 5 min while shaking at 500 r.p.m. on a thermomixer. The DMS was neutralized by adding 60 μl β-mercaptoethanol (Millipore-Sigma). The RNA was purified using RNA Clean and Concentrator −5 kit.

Both linker and target-based DMS-MaPseq library generation protocol as described above can be used after this step.

**Ex virio RNA extraction and DMS modification.** Total SARS-CoV-2 RNA was extracted from the supernatant of infected Vero cells (as described above), resuspended in 1 ml TRIzol (ThermoFisher Scientific) and RNA was extracted following the manufacturer's specifications. The RNA was purified using RNA Clean and Concentrator −5 kit (Zymo) and DMS modified as described in "In vitro transcribed FSE and DMS modification", For the FSE regions a target-based DMS-MaPseq library generation protocol was used using primers P4 and P5 (Supplementary Data 5).

**Dual-luciferase frameshift reporter assay.** FSE-containing segments of 92 nt and 2924 nt (which correspond to nucleotides 13,457–13,548 and nucleotides 12,686–15,609, respectively, based on 2019-nCoV/USA-WA1/2020) were inserted into dual-luciferase reporter plasmid pmirGLO (Promega) between firefly luciferase (Fluc) coding sequence in the 0 frame and Renilla luciferase (Rluc) coding sequence in the −1 frame. Insertion of 0-frame stop codon between Fluc and the FSE is used as negative control construct while a construct of matching length in which Fluc and Rluc were translated continuously without frameshifting is used as a positive control.

Frameshifting reporter as well as positive and negative control mRNAs were in vitro transcribed and polyadenylated using HiScribe T7 mRNA kit (New England Biolabs) and capped using the Vaccinia Capping System (New England Biolabs), and a poly(A) tail was added using *E. coli* Poly(A) Polymerase (New England Biolabs) according to the manufacturers' instructions. Purified mRNAs were transfected in HEK293T cells in 24-well plates using Lipofectamine MessengerMAX (ThermoFisher). Twenty-four hours after transfection, cells were washed once with phosphate-buffered saline (PBS), and lysed in Glo Lysis Buffer (Promega) at room temperature for 5 min. Ten microlitres of lysate was diluted with 30 μL PBS before being mixed with 40 μL Dual-Glo Fluc substrate (Promega). After 10 min, Fluc activity was measured in a GloMax 20/20 luminometer (Promega). Subsequently, 40 μL Dual-Glo Stop & Glo reagent was added to the mixture, incubated for 10 min, and measured for Rluc luminescence. The ratio between Rluc and Fluc activities minus the negative control background luminescence and normalized to positive control luminescence was calculated as frameshift efficiency.

**Mapping and quantification of mutations.** FASTQ files were trimmed using TrimGalore (github.com/FelixKrueger/TrimGalore) to remove Illumina adapters. Trimmed paired reads were mapped to the genome of SARS-CoV-2 isolate SARS-CoV-2/human/USA/USA-WA1/2020 (GenBank: MN985325.1)[52] using Bowtie2[53] with the following parameters: --local --no-unal --no-discordant --no-mixed -L 12 -X 1000. Reads aligning equally well to more than one location were discarded. SAM files from Bowtie2 were converted into BAM files using Picard Tools SamFormatConverter (broadinstitute.github.io/picard).

For each pair of aligned reads, a bit vector of the length of the reference sequence was generated using DREEM[24]. Bit vectors contained a 0 at every position in the reference sequence where the reference sequence matched the read, a 1 at every base at which there was a mismatch or deletion in the read, and no information for every base that was either not in the read or had a Phred score <20. We refer to positions in a bit vector with a 0 or 1 as "informative bits" and all other positions as "uninformative bits."

For each position in the reference sequence, the number of bit-vectors covering the position and the number of reads with mismatches and deletions at the position was counted using DREEM. The ratio of mismatches plus deletions to total coverage at each position was calculated to obtain the population average DMS reactivity for each position.

**Filtering bit vectors.** In cases indicated below, bit vectors were discarded if they had two mutations closer than 4 nucleotides apart, had a mutation next to an uninformative bit, or had more than an allowed total number of mutations (greater than 10% of the length of the bit vector and greater than three standard deviations above the median number of mutations among all bit vectors). The DMS reactivity for each position was computed from the filtered bit vectors in the same way as described above.

**Computing genome coverage and DMS/SHAPE reactivity correlations.** Genome-wide coverage (Fig. 1b) was computed by counting the number of unfiltered bit vectors from the in-cell library that contained an informative bit (0 or 1) at each position. Signal and noise plots (Fig. 1c) were generated from the unfiltered population average mutation rate. The signal and noise were computed every 100 nt, starting at nucleotide 51. For each of these nucleotides, the average mutation rate was computed over the 100 nt window starting 50 bases upstream and ending 49 bases downstream. The "signal" was defined as the average mutation rate of A and C, while the "noise" was defined as the average mutation rate of G and U.

The Pearson correlation of DMS reactivities between biological replicates genome-wide (Fig. 1d) was computed using the filtered bit vectors. The top 0.05% most reactive bases in each dataset were considered outliers and excluded from calculations. Only bases for which both datasets had reactivity data were included in the calculations. The Pearson and Spearman correlations of DMS reactivities between different conditions of the FSE (Fig. 4c) and between clusters and cell types for the FSE (Fig. 5b) were computed using the filtered bit vectors. Pearson's and Spearman's correlations were computed with SciPy[54].

For Pearson and Spearman correlations among other genome-wide chemical probing datasets (Fig. S6), the chemical reactivity datasets were obtained from the corresponding literature[21–23]. Only datasets from the same family of chemical probes (DMS or SHAPE/icSHAPE) were compared to each other. The top 0.05% most reactive bases in each dataset were considered outliers and excluded from calculations. Only bases for which both datasets had reactivity data were included in the calculations. Pearson's and Spearman's correlations were computed with SciPy[54].

**Normalizing the DMS reactivities.** For purposes of folding RNA structures using DMS reactivity constraints, DMS reactivities were normalized to a scale of 0–1 as follows. The median was computed among the top 5% of DMS reactivities (except where a different percentage is specified). All DMS reactivities were divided by this median to compute the normalized reactivities. Normalized reactivities greater than 1.0 were winsorized[55] by setting them to 1.0.

**Folding the entire SARS-CoV-2 genome based on Vero DMS reactivities.** The population average DMS reactivities from Vero cells were obtained from the in-cell library reads as described in "Mapping and quantification of mutations" without "Filtering bit vectors". The 29,882 nt genomes of SARS-CoV-2 was divided into 10 segments, each roughly 3 kb, whose boundaries were predicted to be open and accessible by RNAz[27]. For each segment, the population average DMS reactivities were normalized according to "Normalizing the DMS reactivities" using the 1500 most reactive positions for normalization. All values were set without data (including all guanines and uracils) were set to −999 (unavailable constraints). The segment was then folded using the Fold algorithm from RNAstructure[26] with parameters -m 3 to generate the top three structures, -md 350 to specify a maximum distance of 350 nt between paired bases, and -dms to use the normalized mutation rates as constraints in folding. Connectivity Table files output from Fold were converted to dot-bracket format using ct2dot from RNAstructure[26]. The ten dot-bracket structures were concatenated into a single genome-wide structure.

**Folding and clustering regions of the SARS-CoV-2 genome based on Huh7 DMS reactivities.** Bit vectors for each PCR amplicon were generated separately as described in "Mapping and quantification of mutations". For each amplicon, bit vectors were filtered out if they did not pass the criteria in "Filtering bit vectors" or if <95% of their positions were informative. Positions with raw DMS reactivities less than 0.005 were set to zero to remove noise. The bit-vectors were clustered with DREEM using a maximum of $K = 3$ clusters. The number of clusters ($K$) resulting

in the minimum Bayesian Information Criterion (BIC) was chosen as the optimal $K$, as done previously[24].

Clusters were deemed valid if they met four criteria based on the clusters at $K = 2$: (1) at least 100,000-bit vectors passed the filtering step, (2) the maximum DMS reactivity in either cluster was no more than 0.30, (3) the coefficient of determination ($R^2$) between the DMS reactivities (ignoring zero-valued reactivities) in the two clusters was no more than 0.5, and (4) the ratio of the maximum DMS reactivity in cluster 1 and cluster 2 was between 1/3 and 3. The locations of valid clusters are shown in Fig. 3 (lavender bars).

**Generating receiver operating characteristic curves and computing AUROC.** The AUROC quantifies how well DMS/SHAPE reactivities support the predicted RNA structure, under the assumption that paired bases should be less reactive than unpaired bases. Based on the secondary structure, each position was labeled as paired or unpaired, and the DMS reactivities were partitioned into paired and unpaired groups based on these labels. The ROC curves and AUROC values were computed using SciPy[54]. Here, "true" and "false" positives represent, respectively, paired and unpaired nucleotides with DMS reactivities less than the sliding threshold.

**Computing the mFMI.** The mFMI computes the similarity between two RNA structures from 0 (entirely dissimilar) to 1 (identical). It is common practice[33] to compare two RNA structures by quantifying their shared and unique base pairs using true positive rate (TPR, also called sensitivity) and PPV. The FMI—the geometric mean of TPR and PPV—compresses these values into one similarity score. However, FMI does not account for shared regions of open bases, such that two mostly unstructured RNAs differing by a few base pairs could have a low FMI despite having a similar lack of structure. This modified FMI is a weighted average of the FMI (weighted by the fraction of positions at which at least one RNA has a paired base) and 1 (weighted by the fraction of positions at which both RNAs have an unpaired base). As the amount of structure in both RNAs increases, mFMI approaches FMI; as the number of structures decreases in both RNAs, mFMI approaches 1.

Given two RNA structures of the same length ($L$), all base pairs in each structure are identified. The number of base pairs common to both structures ($P_{12}$) as well as the number of base pairs unique to the first structure ($P_1$) and to the second structure ($P_2$) were computed. Given these quantities, the FMI is defined as $FMI = P_{12}/\sqrt{(P_{12} + P_1)(P_{12} + P_2)}$ [33]. In the case that $(P_{12} + P_1)(P_{12} + P_2) = 0$, we let $FMI = 0$. We then compute the fraction of positions at which both RNA structures have an unpaired base and call it $u$. Then mFMI is defined as $mFMI = u + (1 − u) \times FMI$.

When comparing regions of larger RNA structures, some base pairs may involve one base in the region of interest and the other outside of that region. Such base pairs were treated the same as pairs where both bases were in the region of interest.

**Genome-wide computation of AUROC and mFMI.** Local values of AUROC (Supplementary Fig. 3) and mFMI (Supplementary Fig. 4) were computed using sliding windows of 80 nt in increments of 1 nt. For AUROC, nucleotides with missing reactivity data were ignored, as were Gs and Us for DMS-MaPseq datasets. AUROC was not computed for any window with reactivities on fewer than 5 paired or 5 unpaired bases.

**Benchmarking of AUROC on decoy structures.** We obtained a ground truth secondary structure of the U4/U6 snRNA by processing an NMR structure (2N7M[56]) from the Protein Data Bank[57] with RNApdbee 2.0[58]. We obtained ground truth secondary structures for the 5-stem and 4-stem structures of HIV-1 RRE from a previous study using SHAPE[59]. For each RNA structure, we generated decoy structures in two ways: (1) refolded the RNA sequence using RNAstructure's Fold algorithm[60] without chemical probing constraints at a temperature of 273.15 K (to generate a set of suboptimal structures) and (2) eliminated base pairs randomly from each ground truth structure to generate a set of 1000 additional decoys. For each decoy structure, we computed the AUROC with respect to our previously collected DMS-MaPseq data[24] and computed the mFMI with respect to the ground truth structure.

**Folding the FSE from Vero cell data.** Reads from RT-PCR of a 283 nt segment of in-cell RNA spanning the FSE (nucleotides 13,342–13,624) were used to generate bit vectors. The bit-vectors were filtered as described above, and the filtered average mutation rates were normalized. The RNA was folded using the ShapeKnots algorithm from RNAstructure[61] with parameters -m 3 to generate three structures and -dms to use the normalized mutation rates as constraints in folding. All signals on G and U bases were set to −999 (unavailable constraints). Connectivity Table files output from Fold were converted to dot-bracket format using ct2dot from RNAstructure[26].

**Folding the FSE from Huh7 cell data.** Using the same bit vectors as described in "Folding and clustering … Huh7 DMS reactivities", we repeated clustering and

folding for the amplicon spanning the FSE region (nucleotides 13,434–13,601). In order to check for long-distance interactions, we folded a sequence containing an additional 1500 nt upstream and downstream of the FSE (coordinates 11,934–15,101) using the Fold algorithm from RNA structure with the DMS reactivity constraints only on As and Cs in the region 13,434 – 13,601, no maximum base pair distance, and -m 3 to generate three structures.

**Coronavirus sequence alignments.** Accession numbers of curated SARS- and MERS-related coronavirus genomes were obtained from[45] and downloaded from NCBI. The sequences were aligned using the MUSCLE[62] web service with default parameters. The region of the multiple sequence alignment spanning the two sides of Alternative Stem 1 was located and the sequence conservation was computed using custom Python scripts.

For the alignment of all betacoronaviruses with genomes in NCBI RefSeq[46], all reference genomes of betacoronaviruses were downloaded from RefSeq using the query "betacoronavirus[organism] AND complete genome" with the RefSeq source database as a filter. The sequences were aligned using the MUSCLE[62] web service with default parameters. The subgenus of betacoronavirus to which each virus belonged was obtained from the NCBI taxonomy database[63].

**Detecting alternative structures genome-wide in Vero cells.** The reference genome (length = 29,882 nt) was partitioned into 373 regions of 80 nt each and one final region of 42 nt. For each region, reads were filtered out if they did not pass the criteria in "Filtering Bit Vectors" or if they did not overlap with at least 20% (16 nt) of the region. The reads were then clustered using the EM algorithm implemented previously[24] using a maximum of two clusters per region, ignoring G and U residues, and setting all mutation rates less than 0.005 to 0.0.

After clustering, regions were filtered out if fewer than 100,000 reads mapped to the region ($n = 42$) or if either cluster contained a nucleotide with a mutation rate exceeding 30% ($n = 16$). For each remaining region with two clusters ($n = 316$), each cluster's DMS reactivities ($\mu$) on As and Cs were normalized by setting the highest reactivity to 1.0 and scaling the reactivities of all other bases proportionally. For each nucleotide, the difference in DMS reactivities ($\Delta$DMS) between its mutation rate in cluster 1 ($\mu_1$) and cluster 2 ($\mu_2$) was calculated as $\Delta$DMS $= |\mu_1 - \mu_2|$.

To identify regions better explained by structural ensembles than by a single population average structure, we first smoothed the $\Delta$DMS values by taking the median over a sliding window of 80 nt in increments of 1 nt, leaving blank any windows containing fewer than 10 informative $\Delta$DMS values on As and Cs. We first identified all positions at which smoothed $\Delta$DMS was greater than its global median (0.162) and local AUROC (see "Genome-wide computation of AUROC and mFMI") was less than its global median (0.897). This process yielded a bit vector corresponding to meeting the aforementioned criteria or not (Fig. 3, gray shading). To remove noise from this vector and guarantee that all contiguous regions would be at least $x$ nt long, we used an iterative process in which we first applied a convolutional low-pass filter that computed, for each nucleotide, the average value of the bits in the window from $x$ nt upstream to $x$ nt downstream, then generated a new bit vector by setting all convolved values less than 0.5 to zero and all greater than 0.5 to 1, and iterated these steps until the bit vector no longer changed (Fig. 3, green bars). We chose $x = 34$ because this length is one smaller than the median length of a structural element (before filtering) in the Vero model of the genome (see "Analyzing covariation … genome structure").

**Analyzing covariation among paired bases in the SARS-CoV-2 genome structure.** Our strategy for analyzing covariation involved breaking the genome-wide structure into individual elements and analyzing each element separately to make the problem computationally tractable. For each element, we built a covariance model, computed a structure-aware multiple alignments with a database of coronavirus sequences, refined the model by repeating the build and align steps three times, and analyzed covariation in the refined alignment.

First, we defined a structural element as a set of contiguous nucleotides in which every nucleotide lies between two nucleotides that form a base pair (including the outermost pair). Intuitively, a structural element is anything that protrudes from the main horizontal line of unpaired bases in the secondary structure diagram (Supplementary Fig. 1). Our model of the population average structure of the SARS-CoV-2 genome from Vero cells contained 456 structural elements with a median length of 35 nt. To reduce ambiguity during the alignment step, we discarded every element whose corresponding sequence was more than 70% identical to any other subsequence of the same length in our SARS-CoV-2 reference genome. This process yielded 353 structural elements encompassing 74.5% of the genome and ranging in length from 12 to 348 nt, with a median length of 42 nt.

To generate our database of coronavirus genome sequences, we downloaded all 369,870 complete coronavirus genomes from GenBank[64] on 2021-07-04 using the following NCBI E-utilities command:

esearch -db nuccore -query 'coronavirus[Organism] AND "complete genome"' | efetch -format fasta > CoVs_NCBI_210704.fasta

We removed all but one copy of every set of identical sequences, yielding a database of 301,535 non-redundant full-length coronavirus genome sequences.

For each structural element, we used the following procedure, based on Infernal, to identify covarying bases. First, build and calibrate a covariance model from a Stockholm alignment file containing only the sequence and structure of the element from our genome-wide model, using cmbuild and cmcalibrate from Infernal[37]. Then, search for homologs in the sequence database with an E-value of 301.535 (corresponding to a false positive rate of about 0.1%) using cmsearch, and keep only unique homologs with no more than 5% ambiguous bases (e.g., Ns). Align the unique homologs using cmalign, then return to the build and calibrate step and repeat the whole process a total of three times to refine the alignment iteratively. Finally, identify covarying base pairs using R-scape[36] with the -s option to specify the structure of the covariance model.

**Quantification of minus-strand reads.** Mapped reads from the in-cell library were classified as minus-strand using a custom Python script if they had the following SAM flags[65]: PAIRED and PROPER_PAIR and ({READ1 and MREVERSE and not REVERSE} or {READ2 and REVERSE and not MREVERSE}) and not (UNMAP or MUNMAP or SECONDARY or QCFAIL or DUP or SUPPLEMENTARY).

**Visualizing RNA structures.** RNA structures were drawn using VARNA[66]. The bases were colored using the normalized DMS signals.

**Reporting summary.** Further information on research design is available in the Nature Research Reporting Summary linked to this article.

## Data availability

The data supporting the findings of this study are available from the corresponding authors upon reasonable request. The short-read sequencing data generated in this study have been deposited into NCBI Gene Expression Omnibus (GEO) under accession code GSE153851. Whole-genome secondary structure models of SARS-CoV-2 in Vero and Huh7 cells are provided in Supplementary Data 6 and 7, respectively. Raw DMS reactivities for each biological replicate of Vero, Vero (aggregated replicates), and Huh7 are provided in Supplementary Data 8. A file of Source Data for all main and supplementary figures is provided with this paper. Source data are provided with this paper.

## Code availability

The source code for the data processing and analyses with DREEM is available at http://dreem.wi.mit.edu/static/dreem.zip and http://dreem.wi.mit.edu/static/DREEM_Manual.pdf. The source code for data analysis and figure generation is available at https://github.com/matthewfallan/SARS-CoV-2_genome_structure and https://github.com/matthewfallan/rouls.

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

## Acknowledgements

The SARS-CoV-2 starting material was provided by the World Reference Center for Emerging Viruses and Arboviruses (WRCEVA), with Natalie Thornburg (nax3@cdc.gov) as the CDC Principal Investigator." We thank T.B. Faust for manuscript input and E. Smith for illustrator images. This work was supported by The Pershing Square Foundation, S.R., The Office of Naval Research Award # N00014-20-1-2084, M.B. and the Burroughs Wellcome fund, S.R.

## Author contributions

T.C.T.L. and S.R. conceived and designed the project. T.C.T.L. carried out all experiments with collaborative contributions from J.Z.W., L.E.M., A.G., C.Z, and A.N. M.F.A., T.C.T.L., S.K., S.S.Y.N., F.Z., and S.R. performed the data analysis. Y.S. and J.U.G. performed the frameshifting reporter assays. T.C.T.L., M.F.A., and S.R. interpreted the results and wrote the paper with input from S.K., S.S.Y.N., L.E.M., M.B., A.G., Y.S., and J.U.G.

## Competing interests

The authors declare no competing interests.
