## [Peer Review File · Nature Communications]

Title: Secondary structural ensembles of the full SARS-CoV-2 RNA genome in infected cellsREVIEWER COMMENTS

Reviewer #1 (Remarks to the Author):

The paper entitled “Insights into the secondary structural ensembles of the full SARS-CoV-2 RNA genome in infected cells” reports an in vivo average structure model of SARS-CoV-2 in infected cells by DMS-MaPseq. The authors have done extensive comparison with some existing models both experimentally inferred or computationally predicted. With the help of the DREEM algorithm, the population average structure data was further deconvoluted into different clusters, suggesting there exists different secondary structure conformations for many regions of SARS-CoV-2, including the FSE region. This study suggests a model distinct from the one proposed by in vitro studies on minimal FSE. Using a reporter system, they found that FSE in the 3kb original sequence context promotes a higher frameshifting rate (~40%) than the minimal FSE (20%). The study directly addresses the RNA structural landscape of SARS-CoV-2 in infected cells and provided a very valuable resource for further mechanistic studies of the molecular virology of SARS-CoV-2. The study is surely timely and of very high significance considering the COVID-19 pandemic. That being said, quite a few issues need to be addressed.

Major:

1. The first concern is about the physiological relevance of the RNA structural data for SARS-CoV-2. The study used a monkey cell as host and the viral load is so high (~40% of total cellular RNA from their reported sequencing data. By the way, the writing reads a little uncertain whether here the “cellular” RNA includes the viral fraction) that it raises the concern how the structural profiles could represent those in human cells. What about the cytopathic effects of the cells in the study? Studies have been published using human cells as host, including one from Mol. Cell (Huston et al 2020), one from Cell (Sun et al 2021). Comprehensive comparisons should be performed to check the agreement with those studies. As the authors claimed, the advantage of this study is the devolution of structural ensembles, which has not been accomplished. The strength of this study will be improved if the authors found agreement for the structural probing data between this study and those in human cells.
2. The second big concern is about the DREEM approach. One-hit chemistry is vital for probing RNA structures in its native conformation. Every introduced modification could potentially perturb the target RNA structure, and thus multiple modifications should be avoided in RNA structure probing. However, for the DREEM approach to work, a sequencing read (aka, an RNA fragment) should be modified for multiple times. For sure, the value in doing this is to help deconvolve structural ensembles. However, careful control should be done to ensure that excessive modification will not drastically affect the target RNA structures. Ideally a titration of different modification conditions with comparison on resulting structure could help alleviate the concern. At least, for the intense modifications used here, their effects on the structures of RNAs of different expression levels should be quantified, the average modification frequency (and not every modification will lead to a reverse transcription mutation!) should be estimated, warning should be included for the readers.
3. The use of DSCI to support the validity of the proposed models is definitely a circular reasoning. The structure model is built with probing data as constraints at the first place, and then use the agreement between the model and the probing data as a way to assess the quality of the model?

4. Co-evolution has been used in many RNA structural analyses to support the structural models built. Are there any co-evolution supports for the models built in this study? This reviewer is even more interested in whether the authors can identify some co-evolution supports for alternative structures. That'll be very interesting.
5. The authors showed the agreement between the structural models built in this study and the models from previous studies, but only for the 5'utr of SARS-CoV-2. What about 3'utr? What about the cellular RNAs?
6. Lines 241~243, this reviewer would like to note that the intercept and slope are probably more importantly two parameters than "1) the maximum allowed distance for base pairing and 2) the threshold for DMS signal normalization.". The authors should explicitly give the information of the two parameters and justify their choice.
7. Why are so many regions of Δ DMS empty in Figure 3 and some of them are of big length? By the way, it labels Δ mut in Figure 3, is this the same as Δ DMS?
8. It is better to quantify the Δ DMS between loops and stems within the regions classified into high DSCI and high Δ DMS rather than a statement without supporting quantification.
9. The very long section of alternative structure discussion needs more display items. The results here include that 316 of the 373 regions (lines ~369-373) formed at least two clusters. Exactly how many? How distinct between the clusters? The authors should use figures to show these results. And, for each region, how was the number of clusters chosen? And how to evaluate the cluster result is reasonable or biological relevant rather than artifacts?
10. And it is interesting that the high DSCI and high Δ DMS regions exist. Since the Δ DMS is high between two clusters, I am curious about how well the population average structure is supported by the reactivities of each cluster estimated in DSCI? Is it supported by the reactivities of one cluster or both or neither? Also, here it needs display items for the example (lines ~395-396). The same for ~lines 416-417.
11. The average structure data doesn't generate high DSCI structure for FSE region and the clustered data also does not. Is it because the clustering result of mutation pattern is not correct or there exists more than two clusters?
12. The alternative structure AS1 is confusing. Is this one of multiple alternative structures? If yes, then what are the others? In addition, the assertion of the lack of the pseudoknot is not very convincing. It is possible that the pseudoknot is very flexible. Mutation or co-evolution studies may help further gauge the probability of its existence (such analyses are also suggested for AS1).
13. Figure 5, this whole paper emphasizes a lot on the structural ensemble. And here Figure 5A shows the two DMS profile clusters, yet the structural model in Figure 5B remains an average one. I would hope to see the different structural models derived from two DMS profiles in Figure 5A.
14. The alternative structure (or structural ensemble) analyses are apparently the major contribution to from this study. However, all analyses are based on one single tool. Is it possible for the authors to compare their results with crosslinking (Ziv et al 2020, Wan et al 2021) and nanopore sequencing (Yang et al 2021) based methods?
15. The ribosome frameshifting experiment is beautiful. However, first, importantly, the structures of the FSE in the two constructs should be confirmed. Second, to make the experiments more convincing in support of the hypothesis, it would be nice to also include additional experiments and controls. For example, what about to include the whole 283 nt fragment tested in Figure 4?

16. This reviewer also suggests to delete or at least tone down many claims of “first” throughout the manuscript.

Minor:

1. On line 334, citation of Figure 2B should be Figure 2D
2. On lines 262 and 286, dangling citations to Figure 3A and Figure 3C
3. Figure 5A, position 152 is A, but in figure 5B, it is U. Why?

Reviewer #2 (Remarks to the Author):

This manuscript presents the results of extensive experimental probing of the SARS-CoV-2 RNA structurome. The authors combine in vitro and in cellulo probing of SARS-CoV-2 using DMS-MaPseq in conjunction with their unique DREEM data analysis pipeline to define a genomic secondary structural map for SARS-CoV-2, as well as indicate regions of the viral genome that are prone to forming dynamic/alternative conformations. This important work adds distinctive layers of information onto a growing body of results that define RNA secondary structure of SARS-CoV-2. A key finding was that over 80% of the viral genome has evidence of forming alternative structures; they go on to detail a dramatic shift in structure for the viral frameshifting stimulation element (FSE) that has functional implications to a key viral regulatory process needed to express essential genes. Interestingly, they show a sequence context dependence of the FSE structure that reiterates the importance of considering the in vivo (cellulo) conditions and native sequence context of an RNA structure. Their results have great significance toward efforts at therapeutic targeting of viral RNA, as well as many implications to basic biology, and should be of great interest to a wide array of researchers. The manuscript is well-crafted and would make a great addition to Nat. Comm. after addressing the comments described below.

Major Comments:

1. My major critique is that one of the key results reported here, that >80% of SARS-CoV-2 genome structure may have alternative structures, is not explored more. Not every structural cluster needs to be treated with the same detail as their analysis of the FSE, but some additional information on these other regions would be very helpful.

1A. Of the 80% of the genome that has evidence of structural dynamics, can the authors estimate the number of structural clusters and approximate abundance of each conformation? It is also critical to make these models easily available to readers in a format that facilitates their analysis and usage.

1B. How different are the conformations from each other: e.g., what is the % similarity between alternative conformations vs. the population average DMS model (perhaps reporting the mFMI metric)?

1C. To address the issues above, a table defining the genomic coordinates of each identified region, the number of alternative conformations, their abundance, and similarity to the population average model would be very helpful; particularly if these entries are linked some way to the structural models (in dot-bracket or ct format; perhaps even embedded in table). In this way, readers can easily assess the potential significance of an alternative fold as well as access the model structures.

2. A second concern regards some clarifications that could be helpful in the Methods section:

The first section “RT-PCR and sequencing of DMS-modified RNA” describes the RT process for DMS modified RNA and subsequent purification of cDNA product. Following this is a description of PCR amplification, library prep with the NEBNext Ultra kit, and iSeq 100 sequencing.

The next section “Library generation with DMS-modified SARS-CoV-2 RNA” describes the process of fragmenting RNA, purification/size selection of RNA, fragment end repair, and ligation of a linker sequence. Following this is a description of another (or the same?) RT process. Resulting cDNA is size selected and circularized and subsequent PCR amplification introduces sequencing adapters for iSeq 100 sequencing.

2A. As a reader, it is unclear (even after referencing Zubradt et al. 2016) whether these two sections are describing separate library generation protocols that were each completed (as the RT process is repeated twice) or if the second section is going into greater detail about steps taken during library generation in the first section. If they are describing the same process, perhaps integrating these sections would make things more clear and less redundant. If they are indeed separate process, perhaps some additional explanation is required to help make that distinction.

2B. A list of primers used in the study would help to lend clarity to the process and be informative to the research community.

Minor Comments:

1. The sentence on line 878-879 “to produce either 150x150-nt paired-end reads.” is somewhat confusing.

2. In some places “nt” were appended to numbers with no spaces and elsewhere, spaces were included between the two: e.g. line 883, where there is also an instance where “nucleotides” is spelled out and prepended to genomic coordinates with no space.

-Walter Moss

Reviewer #3 (Remarks to the Author):

This article uses DMS-MaPseq to report the secondary structure ensembles of the entire SARS-CoV-2 genome in infected cells. On this basis, the author reconstructed and compared the in vivo and in vitro FSE secondary structure; observed the variable structure in FSE, and quantified the frameshifting rates promoted by this conformation.

In terms of novelty, this article is not the first one to report the secondary structure of the full-length genome of SARS-COV2 based on single-stranded RNA capture method. The authors reconstructed the variable structure of FSE, which is consistent with the known NMR results, and detected the ratio of different variable structures, which belongs to the verification of the known information. Compared with published articles, in addition to the limited improvement in the accuracy of the secondary structure (it may also be due to the inappropriate QC method or circular argumentation, or the lack of distinction between genomes and subgenomes RNA, which result in low data quality and depressed accuracy of reconstruction), there is no other information provided. In conclusion, this article is not suitable for publication in Nature Communications in my opinion. The following are the major comments and some mistakes in this manuscript, hoping to be helpful for authors.

Critical comments:

1. Line 158, authors state "...has been shown to yield structures of similar or slightly higher accuracies compared to SHAPE". How similar is the DMS-seq signal generated by this project to the SHAPE signal in (Sun et al., Cell, 2021 and Huston et al., Mol Cell. 2020)?
2. In the reconstructed structure (Supplementary Figure 1), how many base pairs are consistent or inconsistent compared with Sun's data set? What is the improvement compared with this article? What about Ziv et al. (Mol Cell. 2020; PMID: 33259809)? As it is unfair to evaluate their structure by using your data or to evaluate your structure by using their data, the Psoralen enriched PCR validation is needed for the different base pairings between this project and published articles in full-length virus genome from purified RNA sample (Psoralen enriched PCR: Dadonaite et al., Mat Microbiol., 2019; PMID: 31332385).
3. What is the ratio of the virus full-length genome RNA and subgenome RNAs (sgRNAs) in DMS-seq? In Fig1C, the DMS-seq signal coverage shows that the subgenome is the main component in the data. How to distinguish whether the DMS-seq signal comes from the full-length genome or sgRNAs? The DMS-seq signal after 20kb and before 7kb will be affected by the canonical sgRNAs and non-canonical sgRNAs respectively (Fig1 and FigS1 in Kim et al., Cell, 2020; PMID: 32330414; Wang, Dehe et al. Molecular Cell, 2021). The experimental strategy used in Ziv et al. (Mol Cell. 2020) to distinguish full-length genomes and sgRNAs should be used.
4. Line 199, the difference of data sets between this project and other article may be just caused by the different proportion of the sgRNAs, as the structure of sgRNAs is not the same as whole genome (Sun et al., Cell, 2021). Comparison with his own data set is unfair, unless proving that this DMS-seq data set is much better than others. Comparing the structures in different data sets can help to prevent circular argumentation. The SHAPE-score distribution and corresponding DSCI value also need to be calculated in their data set. (Sun et al., Cell, 2021 and Huston et al., Mol Cell, 2020)
5. It is also inappropriate for the author to use DSCI to evaluate the quality of the secondary structure. For example, A, B, C and D are paired regions, and E, F are unpaired regions. If the real base-pairing situation is A::B / C::D and the predict result is A::D / B::C, the predict structure is obviously wrong but

the DSCI score is same as the real one.

6. How many alternative structures in SARS-CoV-2 genome / sgRNAs?

7. The results that the longer FSE promoted more frameshifting compared with the short FSE were easily predicted, but more structure details about the mechanism to regulate frameshifting should be studied with mutation or truncation.

Other comments:

1. DMS-MapSeq is not the method created in this article. The content of the detailed experimental process for DMS-MapSeq method described in Fig1A should be removed.

2. What does points represent in Fig1B? One nucleotide or a genome region? For non-normal samples, Spearman's correlation coefficient should be used instead of Pearson's correlation coefficient.

3. Line 199, how about the mutation distribution and DSCI value of the RNA structures predicted by purely computational models like RNAfold and RNAstructure? And how about the structure in Sun's article? Comparing these structures in one graph helps to show the correctness.

4. Also, using different types of public data set like PARIS and COMREADS to check your structure, especially long-range interactions, is helpful.

5. In Fig 2C, the statistical range should be marked on the figure like Fig 2A.

6. The illustrator in Fig 4B is incomprehensible.

7. No figure 3A, 3C were found in this manuscript.

Below we include a point-by-point response to the reviewer comments.

Reviewer #1 (Remarks to the Author):

The paper entitled “Insights into the secondary structural ensembles of the full SARS-CoV-2 RNA genome in infected cells” reports an in vivo average structure model of SARS-CoV-2 in infected cells by DMS-MaPseq. The authors have done extensive comparison with some existing models both experimentally inferred or computationally predicted. With the help of the DREEM algorithm, the population average structure data was further deconvoluted into different clusters, suggesting there exists different secondary structure conformations for many regions of SARS-CoV-2, including the FSE region. This study suggests a model distinct from the one proposed by in vitro studies on minimal FSE. Using a reporter system, they found that FSE in the 3kb original sequence context promotes a higher frameshifting rate (~40%) than the minimal FSE

(20%). The study directly addresses the RNA structural landscape of SARS-CoV-2 in infected cells and provided a very valuable resource for further mechanistic studies of the molecular virology of SARS-CoV-2. The study is surely timely and of very high significance considering the COVID-19 pandemic. That being said, quite a few issues need to be addressed.

Major:

1. The first concern is about the physiological relevance of the RNA structural data for SARS-CoV-2. The study used a monkey cell as host and the viral load is so high (~40% of total cellular RNA from their reported sequencing data. By the way, the writing reads a little uncertain whether here the “cellular” RNA includes the viral fraction) that it raises the concern how the structural profiles could represent those in human cells. What about the cytopathic effects of the cells in the study? Studies have been published using human cells as host, including one from *Mol. Cell* (Huston et al. 2020), one from *Cell* (Sun et al. 2021). Comprehensive comparisons should be performed to check the agreement with those studies. As the authors claimed, the advantage of this study is the devolution of structural ensembles, which has not been accomplished. The strength of this study will be improved if the authors found agreement for the structural probing data between this study and those in human cells.

Previous studies on the genome structure used monkey cells or human cells, but no single study used one chemical to compare these cell types. As we discuss above, the two studies that used Vero cells and SHAPE reagent have low agreement with each other (Pearson=0.24, see (1) in main response, also New Supp. Fig. S6). This low agreement under presumably identical conditions in Vero cells precludes the comparison of SARS-CoV-2 structure between human and monkey cells from published work.

In order to address the critical question raised by the reviewer as to how much the cell type and viral load affects the viral RNA folding, we probed the structure of SARS-CoV-2 in infected Huh7 (human) cells. Since the viral load in Huh7 cells is relatively low (~2%), we used specific primers to amplify and sequence just the SARS-CoV-2 genome (updated Figure 1). We find that the structure is largely consistent between human and monkey cells (Pearson=0.84). Importantly, using the same library strategy (specific RT/PCR, which gives the highest signal to noise ratio) for both Vero and Huh7 cells, we find that the RNA structure ensemble at the frameshifting stimulation element is identical (within error of biological reproducibility) in human and monkey cells (Pearson=0.98, updated Fig 4B).

2. The second big concern is about the DREEM approach. One-hit chemistry is vital for probing RNA structures in its native conformation. Every introduced modification could potentially perturb the target RNA structure, and thus multiple modifications should be avoided in RNA structure probing. However, for the DREEM approach to work, a sequencing read (aka, an RNA fragment) should be modified for multiple times. For sure, the value in doing this is to help deconvolve structural ensembles. However,

careful control should be done to ensure that excessive modification will not drastically affect the target RNA structures. Ideally a titration of different modification conditions with comparison on resulting structure could help alleviate the concern. At least, for the intense modifications used here, their effects on the structures of RNAs of different expression levels should be quantified, the average modification frequency (and not every modification will lead to a reverse transcription mutation!) should be estimated, warning should be included for the readers.

Absolutely, it is very important to do titration experiments to ensure that the multiple DMS modifications are not changing the RNA structure. Such titrations were performed genome-wide and on specific structures in Rouskin et al. Nature 2014, Zubradt et al. Nature Methods 2017, and Tomezsko et al. Nature 2020. In Zubradt et al. we also estimate that a DMS modification has ~70% chance of getting converted to a mutation by TGIRT (from DMS-like endogenous modifications on the ribosomal RNA). In the current manuscript we do not use higher modification rates than what was used in DMS-seq for mammalian cell treatments (Rouskin et al. Nature 2015, Tomezsko et al. Nature 2020), which was validated on multiple positive control structures.

To further address the reviewer's concern, we specifically performed titration of the 283 nt *in vitro* re-folded frameshifting stimulation element (FSE). As our sequencing length is 300 nt, this experiment allows us to directly count the number of mutations per full length DNA molecule and deduce the number of DMS modifications per RNA molecule. DMS concentrations of 0.1% result in ~1 modification per molecule and DMS concentrations of 10% result on average of ~4 mutations (new Supp. Fig. S12).

Below we show the correlation of the DMS signal at different titrations, revealing that the structure stays the same (within the error of signal reproducibility) between 0.1%-10% DMS (Pearson=0.98). In the current manuscript we use concentrations of 2-3% DMS in cells (that lead to average of 2-3 modifications per 300 nt) and 2-3% *in vitro*.

3. The use of DSCI to support the validity of the proposed models is definitely a circular reasoning. The structure model is built with probing data as constraints at the first place,

and then use the agreement between the model and the probing data as a way to assess the quality of the model?

We thank the reviewer for bringing this point up (similar to Reviewer 3). We didn't clearly explain the metric and why even though it appears as a circular reasoning it is actually not. Validating the results of any structure prediction algorithm is necessary because they depend on the details of the algorithm and its parameters (see point 6 for more details). It is common practice to validate predicted structures by comparing them to the probing data using the area under the receiver operating characteristic curve (AUROC); for example, see Luo et al. (PMC8184798) Fig. S1D and Rouskin et al. (PMC3966492) Fig. 2D.

Three other papers on the SARS-CoV-2 genome structure measure how well their predicted models agree with their own probing data: Sun et al. (Fig. S1C), Manfredonia et al. (Fig. S1), and Huston et al. (Fig. 1D). Sun et al. used AUROC to validate their structures. We have realized that our metric, DSCI, is identical to AUROC, so we have renamed it to the more widely-recognized term AUROC in our revised manuscript (see main point (1) and Fig. 2 of our manuscript).

4. Co-evolution has been used in many RNA structural analyses to support the structural models built. Are there any co-evolution supports for the models built in this study? This reviewer is even more interested in whether the authors can identify some co-evolution supports for alternative structures. That'll be very interesting.

Co-evolution is a great metric however it is challenging in this case because 1) vast majority of the sequence is over coding regions and it is difficult to separate the pressure for conserving amino acids 2) the presence of alternative structures, which could have co-dependent evolutionary pressures.

Despite these challenges, we analyzed covariation using Infernal and R-scape and found support for 63 structures, including known structures in the 3' UTR as well as new structures (new Supplementary Tables 2 and 3).

5. The authors showed the agreement between the structural models built in this study and the models from previous studies, but only for the 5'utr of SARS-CoV-2. What about 3'utr? What about the cellular RNAs?

Thank you for this point, we also included detailed agreement of the 3' UTR (new Supp. Fig. S7). Due to the design of our study, we subtracted cellular non coding RNAs or used SARS-CoV-2 specific primers.

6. Lines 241~243, this reviewer would like to note that the intercept and slope are probably more importantly two parameters than "1) the maximum allowed distance for

base pairing and 2) the threshold for DMS signal normalization.”. The authors should explicitly give the information of the two parameters and justify their choice.

The slope and intercept parameters are part of one common approach to constrain structure predictions with probing data that uses a pseudoenergy function to convert the SHAPE/DMS reactivity of each paired base into an energy score (Deigan et al. 2009, PMC2629221). In this method, the slope and intercept parameters must be empirically derived; this method is used by Sun et al. and Huston et al. However, we instead use a more direct probabilistic energy function (Cordero et al. 2012, PMC3448840) that does not use slope and intercept parameters but rather computes an energy score from the log-odds that each base is paired given its DMS/SHAPE reactivity. We have now clarified this in the methods (lines 777-798).

7. Why are so many regions of Δ DMS empty in Figure 3 and some of them are of big length? By the way, it labels Δ mut in Figure 3, is this the same as Δ DMS?

We apologize for the misunderstanding; the “empty” regions are simply not shaded meaning they were not identified as heterogeneous. We hope this is clearer now in the updated Figure 3.

Yes, Δ mut is the same as Δ DMS; we have changed it all to Δ DMS.

8. It is better to quantify the Δ DMS between loops and stems within the regions classified into high DSCI and high Δ DMS rather than a statement without supporting quantification.

Yes in principle, but since it’s a population average model, we rather not classify a stem as a stem because it could be a loop in the alternative structure.

9. The very long section of alternative structure discussion needs more display items. The results here include that 316 of the 373 regions (lines ~369-373) formed at least two clusters. Exactly how many? How distinct between the clusters? The authors should use figures to show these results. And, for each region, how was the number of clusters chosen? And how to evaluate the cluster result is reasonable or biological relevant rather than artifacts?

Thank you for this important feedback. The DREEM algorithm works by first separating the most distinct clusters and those with the highest abundance. We use stringent criteria to prevent data overfitting - Bayesian Information Criteria (BIC). However, as the reviewer eludes the BIC alone is not sufficient to distinguish between a biologically relevant cluster and an artifact. Although all our simulated data does not over-cluster (Tomezsko et al. 2020), it is possible that artifacts can arise during the library generation strategy or sequencing. However, because there are so few “ground truth”

controls for alternative RNA structures (especially RNAs with more than two conformations), it is indeed non-trivial to determine whether an extra cluster that passes the BIC is biologically relevant. We make the following assumptions for a biologically relevant cluster: 1) under the conditions used no true DMS modification can reach signal greater than 30% mutation 2) two different structures will have similar DMS hit rate such that max DMS signal between them should be within three fold 3) the DMS signal of two different structures will have a low Pearson correlation ($R^2 < 0.5$). These assumptions are based on ground truth experiments used to develop and validate DREEM (Tomezsko et al. 2020).

We have now added a table that gives the coordinates of every region that was identified as heterogeneous in Huh7 cells (Supplementary Table 4), and included the number of clusters that are passing Bayesian Information Criteria (BIC) and the filter for a bona fide structure change described above. We have also clarified these filters in the method section and figure legends (lines 802-807).

10. And it is interesting that the high DSCI and high Δ DMS regions exist. Since the Δ DMS is high between two clusters, I am curious about how well the population average structure is supported by the reactivities of each cluster estimated in DSCI? Is it supported by the reactivities of one cluster or both or neither? Also, here it needs display items for the example (lines ~395-396). The same for ~lines 416-417.

This DSCI (or AUROC) is a very coarse metric and two different structures can both have relatively high DCSI. We are very interested in making this metric more sensitive but this is outside the scope of the current manuscript. We have now added more display items in Supplementary Tables 4 and 6.

11. The average structure data doesn't generate high DSCI structure for FSE region and the clustered data also does not. Is it because the clustering result of mutation pattern is not correct or there exists more than two clusters?

This is correct, which is why originally we did not present a structure model for the clustered data. In principle, both of the reviewer's points are possible, but in this case the DSCI (AUROC) is low because the constrained RNAstructure model is suboptimal. We now reveal a much more accurate RNAstructure model for the alternative clusters in updated Figure 5, which includes a very long-distance RNA:RNA interaction (~1.2 kb away). The clustering result is extremely robust; DREEM converges to the same signal across ten different runs that initiate at random parameters; the clusters pass all filters including BIC and filter for a bona fide structure change (described in point 9). The result is also identical in biological replicates as well as between infected Huh7 and Vero cells (updated Figure 5).

12. The alternative structure AS1 is confusing. Is this one of multiple alternative structures? If yes, then what are the others? In addition, the assertion of the lack of the

pseudoknot is not very convincing. It is possible that the pseudoknot is very flexible. Mutation or co-evolution studies may help further gauge the probability of its existence (such analyses are also suggested for AS1).

Both of the major structures we find in cells are different than the canonical pseudoknot (updated Figure 5). The formation of AS1 in cells has now been supported by multiple studies (*Biochem Soc Trans* (2021) 49 (1): 341–352.). We were able to find the pseudoknot forming in the 3 kb construct as a low abundance cluster *in vitro*. This suggests that the pseudoknot can form transiently in cells and is below the limits of detection (~10%). We have clarified the contribution of the pseudoknot in the discussion (lines 446-456)

13. Figure 5, this whole paper emphasizes a lot on the structural ensemble. And here Figure 5A shows the two DMS profile clusters, yet the structural model in Figure 5B remains an average one. I would hope to see the different structural models derived from two DMS profiles in Figure 5A.

Yes, thank you for pointing this out. We have now provided the model for the two major alternative structures, revealing the long distance interaction at the FSE region.

14. The alternative structure (or structural ensemble) analyses are apparently the major contribution to from this study. However, all analyses are based on one single tool. Is it possible for the authors to compare their results with crosslinking (Ziv et al. 2020, Wan et al. 2021) and nanopore sequencing (Yang et al. 2021) based methods?

It is not trivial to compare these very different approaches as they capture distinct structural features. Nanopore sequencing (Yang et al.) or crosslinking (Ziv et al.) are not quantitative – i.e. they can find a RNA:RNA interaction but cannot determine whether it happens in 5% of molecules or in 50%. However, in our revised manuscript we point out that cluster 2 at the FSE, which is a highly abundant cluster (~50%), is largely consistent with the long-distance interaction proposed by Ziv et al. (lines 384-390)

15. The ribosome frameshifting experiment is beautiful. However, first, importantly, the structures of the FSE in the two constructs should be confirmed. Second, to make the experiments more convincing in support of the hypothesis, it would be nice to also include additional experiments and controls. For example, what about to include the whole 283 nt fragment tested in Figure 4?

Thank you! We confirmed that the 3 kb FSE folds very similarly in cells in the context of the luciferase construct (Pearson=0.82) and most importantly it is not affected by the addition of luciferase (Person=0.99, Supplementary Figure 12B). We did not test the 283 nt in the luciferase assay because 1) our new experiments in updated Figure 4 and structure model in updated Figure 5 show that we need at least a 3 kb construct to recapitulate the structure ensemble found during infection 2) other labs tested longer constructs for frameshifting but only measured ~ 25% frameshifting rate (e.g. ~700 nt constructs from Bhatt et al. 2021).

16. *This reviewer also suggests to delete or at least tone down many claims of “first” throughout the manuscript.*

Thank you for this suggestion, we have now done this.

Minor:

1. On line 334, citation of Figure 2B should be Figure 2D
2. On lines 262 and 286, dangling citations to Figure 3A and Figure 3C

We fixed this, thank you.

3. *Figure 5A, position 152 is A, but in figure 5B, it is U. Why?*

We corrected this and labeled the figure based on the genomic coordinates to make it easier for the reader.

Reviewer #2 (Remarks to the Author):

This manuscript presents the results of extensive experimental probing of the SARS-CoV-2 RNA structure. The authors combine in vitro and in cellulo probing of SARS-CoV-2 using DMS-MaPseq in conjunction with their unique DREEM data analysis pipeline to define a genomic secondary structural map for SARS-CoV-2, as well as indicate regions of the viral genome that are prone to forming dynamic/alternative conformations. This important work adds distinctive layers of information onto a growing body of results that define RNA secondary structure of SARS-CoV-2. A key finding was that over 80% of the viral genome has evidence of forming alternative structures; they go on to detail a dramatic shift in structure for the viral frameshifting stimulation element (FSE) that has functional implications to a key viral regulatory process needed to express essential genes. Interestingly, they show a sequence context dependence of the FSE structure that reiterates the importance of considering the in vivo (cellulo) conditions and native sequence context of an RNA structure. Their results have great significance toward efforts at therapeutic targeting of viral RNA, as well as many implications to basic biology, and should be of great interest to a wide array of researchers. The manuscript is well-crafted and would make a great addition to Nat. Comm. after addressing the comments described below.

Major Comments:

1. *My major critique is that one of the key results reported here, that >80% of SARS-CoV-2 genome structure may have alternative structures, is not explored more. Not every structural cluster needs to be treated with the same detail as their analysis of the FSE, but some additional information on these other regions would be very helpful.*

Yes indeed. To explore the alternative structures more we collected RT-PCR data from infected Huh7 cells, which is more amenable to clustering because of longer molecules ~250 nt as opposed to ~80 nt fragments from library generation. We identified heterogeneous regions that largely overlapped with the data from Vero cells (updated Fig1 and Fig 3).

1A. Of the 80% of the genome that has evidence of structural dynamics, can the authors estimate the number of structural clusters and approximate abundance of each conformation? It is also critical to make these models easily available to readers in a format that facilitates their analysis and usage.

Thank you for this important feedback. We provide a table with the coordinates of 71 heterogeneous regions identified both in Huh7 cells, number of clusters, abundance for each conformation, cluster similarity, and DMS signal for each cluster (Supplementary Table 4).

1B. How different are the conformations from each other: e.g., what is the % similarity between alternative conformations vs. the population average DMS model (perhaps reporting the mFMI metric)?

Please also see response to Reviewer 1 points (9) and (11). The clustering results are extremely robust but the modeling of these results provides challenges, especially since in the case of SARS-2 there is evidence for abundant long-distance interactions (Ziv et al. 2020 and Yang et al. 2021). Since computational modeling is not well suited for long-distance interactions, we believe it could be misleading to provide models for all the clusters. We have provided all the clustering data for the 71 regions that pass all filters for bona fide structure change and included the DMS signal in an easy-to-use machine parsable format (Supplementary Table 6). This table allows others to tests agreement with our data and any structure model.

1C. To address the issues above, a table defining the genomic coordinates of each identified region, the number of alternative conformations, their abundance, and similarity to the population average model would be very helpful; particularly if these entries are linked some way to the structural models (in dot-bracket or ct format; perhaps even embedded in table). In this way, readers can easily assess the potential significance of an alternative fold as well as access the model structures.

Yes, thank you for pointing this out, we have provided such a table (Supplementary Table 4).

2. A second concern regards some clarifications that could be helpful in the Methods section:

The first section "RT-PCR and sequencing of DMS-modified RNA" describes the RT process for DMS modified RNA and subsequent purification of cDNA product. Following

this is a description of PCR amplification, library prep with the NEBNext Ultra kit, and iSeq 100 sequencing.

The next section “Library generation with DMS-modified SARS-CoV-2 RNA” describes the process of fragmenting RNA, purification/size selection of RNA, fragment end repair, and ligation of a linker sequence. Following this is a description of another (or the same?) RT process. Resulting cDNA is size selected and circularized and subsequent PCR amplification introduces sequencing adapters for iSeq 100 sequencing.

2A. As a reader, it is unclear (even after referencing Zubradt et al. 2016) whether these two sections are describing separate library generation protocols that were each completed (as the RT process is repeated twice) or if the second section is going into greater detail about steps taken during library generation in the first section. If they are describing the same process, perhaps integrating these sections would make things more clear and less redundant. If they are indeed separate process, perhaps some additional explanation is required to help make that distinction.

Thank you for bringing this to our attention, we have now restructured our DMS-MaPseq method description to more clearly distinguish between the different library generation strategies our lab used. One is a fragmentation and linker-ligation strategy followed by custom primer amplification and the other is using tiled target-specific primers followed by the NEB ultra kit to generate a sequencing library. (see lines 669-716) We hope this is clearer now for the reader.

2B. A list of primers used in the study would help to lend clarity to the process and be informative to the research community.

We have added this list as Supplementary Table 5.

Minor Comments:

1. The sentence on line 878-879 “to produce either 150x150-nt paired-end reads.” is somewhat confusing.

2. In some places “nt” were appended to numbers with no spaces and elsewhere, spaces were included between the two: e.g. line 883, where there is also an instance where “nucleotides” is spelled out and prepended to genomic coordinates with no space.

-Walter Moss

Reviewer #3 (Remarks to the Author):

This article uses DMS-MaPseq to report the secondary structure ensembles of the entire SARS-CoV-2 genome in infected cells. On this basis, the author reconstructed

and compared the in vivo and in vitro FSE secondary structure; observed the variable structure in FSE, and quantified the frameshifting rates promoted by this conformation. In terms of novelty, this article is not the first one to report the secondary structure of the full-length genome of SARS-COV2 based on single-stranded RNA capture method. The authors reconstructed the variable structure of FSE, which is consistent with the known NMR results, and detected the ratio of different variable structures, which belongs to the verification of the known information. Compared with published articles, in addition to the limited improvement in the accuracy of the secondary structure (it may also be due to the inappropriate QC method or circular argumentation, or the lack of distinction between genomes and subgenomes RNA, which result in low data quality and depressed accuracy of reconstruction), there is no other information provided. In conclusion, this article is not suitable for publication in Nature Communications in my opinion. The following are the major comments and some mistakes in this manuscript, hoping to be helpful for authors.

We thank the reviewer for the thoughts and comments. Indeed, there is previous work on the structure of SARS-CoV-2 – Manfredonia et al. and Huston et al. – but despite using the same chemical (SHAPE) and the same cell type (Vero cells), the data from these groups has poor correlation ($r = 0.25$, see also point 1 in main response). In fact, there have been concerns as to how well SHAPE works in mammalian cells due to limited permeability, which stipulated the development of ic-SHAPE (Spitale et al.). ic-SHAPE, used by Sun et al., is more cell permeable but has its own biases and may not report accurately on RNA structure (Busan et al. *Biochemistry* 2019). More generally, SHAPE and its derivatives are much bulkier than DMS and are not specific to the Watson-Crick pairing positions that report directly on secondary structure. Instead, SHAPE chemicals react with the 2' OH of the ribose, which is an indirect measurement of base-pairing. Ours is the only study that used DMS in cells. DMS is the oldest and most established chemical for RNA structure mapping; it reports directly on base pairing (accessibility of N1A and N3C).

To address the reviewer's main concerns, we collected a new dataset from human cells (Huh7), and identified and provided data for 71 regions across the genome that exists in at least two or three major conformations highlighting the structure ensembles genome-wide (updated Figure 3, Supplementary Tables 4 and 6). It is important to have a complementary approach to SHAPE such as ours, which reports directly on secondary structure, as well as identifies and quantifies alternative RNA conformations.

In terms of the novelty at the frameshift stimulating element (FSE), we apologize if we didn't make the point clearer – our results from infected cells disagree entirely with the NMR work (now clarified on lines 324-336). We also disagree with the published models on the FSE as they are derived based on population averages and our data clearly indicate at least two major conformations, one of which involves a very long distance interaction ~1.2 kb away that could not be seen by NMR, which has a limit of ~200 nt constructs (updated Figure 4). We functionally validate our findings, which indicate that the long distance interactions in the RNA structure ensemble allow high level of

frameshifting (~40%) in infected cells. Based on these findings we propose an entirely new model for frameshifting (Figure 6).

Finally, we demonstrate that the substantial improvement of our model over published models is not due to lack of distinctions of subgenomic RNAs. If we restrict the analysis to ORF1ab, which can only come from the full-length genomic RNA, we get identical results (Sup Fig 5).

Critical comments:

1. Line 158, authors state “...has been shown to yield structures of similar or slightly higher accuracies compared to SHAPE”. How similar is the DMS-seq signal generated by this project to the SHAPE signal in (Sun et al., *Cell*, 2021 and Huston et al., *Mol Cell*, 2020)?

This is an important question. As we discuss above, although generally known to report on accessible residues, DMS and SHAPE are two very different reagents with different physical and chemical properties, and only DMS is a direct readout of base-pairing. However, as we point out in the main response, the agreement between two different labs using SHAPE under presumably identical conditions is very low ($r=0.25$). We have included Supplementary Figure 6 showing the agreement of signal between all published studies.

2. In the reconstructed structure (Supplementary Figure 1), how many base pairs are consistent or inconsistent compared with Sun’s data set? What is the improvement compared with this article? What about Ziv et al. (*Mol Cell*, 2020; PMID: 33259809)? As it is unfair to evaluate their structure by using your data or to evaluate your structure by using their data, the Psoralen enriched PCR validation is needed for the different base pairings between this project and published articles in full-length virus genome from purified RNA sample (Psoralen enriched PCR: Dadonaite et al., *Mat Microbiol.*, 2019; PMID: 31332385).

When using similar human cell types, in the population average models of SARS-CoV-2 infected Huh7 cells, 11,186 base pairs are consistent and 2,160 base pairs are inconsistent between our model and Sun et al. Our model has AUROC of 0.94 compared to 0.83 for Sun et al. (updated Figure 2). In other words, the models derived from averages have substantial agreement with each other (likely due to using similar RNA folding algorithms). However, we find that a large fraction of the genome exists in multiple conformations, therefore the population average model could be misleading. This is why we specifically identify the regions that exists in one versus multiple conformations (Figure 3, Supplementary Table 4).

Ziv et al. do not provide a genome-wide structure model. The psoralen enriched PCR can only identify existence of interactions, which maybe transient, and it cannot quantify the abundance of alternative conformations. Psoralen crosslinking also has bias for longer stems and presence of A-U pairs (Aw et al. *Mol Cell* 2016).

3. *What is the ratio of the virus full-length genome RNA and subgenome RNAs (sgRNAs) in DMS-seq? In Fig1C, the DMS-seq signal coverage shows that the subgenome is the main component in the data. How to distinguish whether the DMS-seq signal comes from the full-length genome or sgRNAs? The DMS-seq signal after 20kb and before 7kb will be affected by the canonical sgRNAs and non-canonical sgRNAs respectively (Fig1 and FigS1 in Kim et al., Cell, 2020; PMID: 32330414; Wang, Dehe et al. Molecular Cell, 2021). The experimental strategy used in Ziv et al. (Mol Cell, 2020) to distinguish full-length genomes and sgRNAs should be used.*

The canonical sgRNAs share the first 69 nt at the 5'UTR and the sequence between 21.5kb-30kb will be the full-length genome. This means that ~1/3 of the genome sequence comes from mixed molecules including both full length and sgRNA. However, this mixing is not a major source of structure heterogeneity we identify because 1) We find similar rate of alternative structures in the 2/3 of the genome that can only come from full length molecules (including the FSE which can only come from full length) 2) Recent work reports that the structure of the genomic and the subgenomic RNAs are nearly identical (Yang et al. BioRxiv 2021) except near the 69 nt leader sequence (Sun et al).

4. *Line 199, the difference of data sets between this project and other article may be just caused by the different proportion of the sgRNAs, as the structure of sgRNAs is not the same as whole genome (Sun et al., Cell, 2021). Comparison with his own data set is unfair, unless proving that this DMS-seq data set is much better than others. Comparing the structures in different data sets can help to prevent circular argumentation. The SHAPE-score distribution and corresponding DSCI value also need to be calculated in their data set. (Sun et al., Cell, 2021 and Huston et al., Mol Cell, 2020)*

We demonstrate that the substantial improvement of our model over published models is not due to lack of distinctions of subgenomic RNAs. If we restrict the analysis to ORF1ab, which can only come from full length genomic RNA, we get identical results (Supp Fig 5).

5. *It is also inappropriate for the author to use DSCI to evaluate the quality of the secondary structure. For example, A, B, C and D are paired regions, and E, F are unpaired regions. If the real base-pairing situation is A::B / C::D and the predict result is A::D / B::C, the predict structure is obviously wrong but the DSCI score is same as the real one.*

Yes, as we point in response to reviewer 1 point 3, DSCI (which is identical to AUROC) is a coarse metric. However, AUROC is a well-established metric that has also been used by others including Sun et al., and further improvement of this metric is outside the scope of this manuscript.

6. How many alternative structures in SARS-CoV-2 genome / sgRNAs?

We identify 71 heterogeneous regions across the genome (Supplementary Tables 4 and 6).

7. The results that the longer FSE promoted more frameshifting compared with the short FSE were easily predicted, but more structure details about the mechanism to regulate frameshifting should be studied with mutation or truncation.

We very politely disagree that these results were easily predicted, as the vast majority of published work on coronaviruses report 20 - 30% of frameshifting (Kelly et al., PMC7397099; Bhatt et al., PMC8168617; Sun et al., PMC7587830; Plant et al., PMC1110908). We have now added more structure details, including the structure model of the long-distance interaction (updated Figure 5).

Other comments:

1. DMS-MapSeq is not the method created in this article. The content of the detailed experimental process for DMS-MapSeq method described in Fig1A should be removed.

We thank the reviewer for this feedback and have changed the figure to emphasize not the method but rather the use of DMS and the comparison between different cell types.

2. What does points represent in Fig1B? One nucleotide or a genome region? For non-normal samples, Spearman's correlation coefficient should be used instead of Pearson's correlation coefficient

The points represent one nucleotide, we clarified this and added Spearman correlations (updated Figure 1).

3. Line 199, how about the mutation distribution and DSCI value of the RNA structures predicted by purely computational models like RNAfold and RNAstructure? And how about the structure in Sun's article? Comparing these structures in one graph helps to show the correctness.

Yes indeed, we have added the DSCI/AUROC values for all published studies and ours in Figure 2 and compared the DMS and SHAPE signals (Supplementary Figure 6).

4. Also, using different types of public data set like PARIS and COMREADS to check your structure, especially long-range interactions, is helpful.

Thank you for the input, importantly we found that the long-range interaction on the FSE agreed well with that of Ziv et. al. (lines 384-390)

5. In Fig 2C, the statistical range should be marked on the figure like Fig 2A.

We have done this on what is now Supp. Fig. S2.

6. *The illustrator in Fig 4B is incomprehensible.*

We have fixed Fig 4B.

7. *No figure 3A, 3C were found in this manuscript . We fixed that.*

REVIEWER COMMENTS

Reviewer #1 (Remarks to the Author):

The revision has very nicely addressed many of my comments. The new data presented in the revision, e.g., the results from human cells, are very beautiful. The following are some remaining questions/concerns.

It is nice to see the new data in human cells and the agreement with the previous results in monkey cells. But I am a little worried to see the low agreements between all the structures, especially from the other studies. In particular, I would hope to see a decent agreement among those in vivo structures. PCC of 0.2~0.3 is way too low. How exactly the correlation is calculated? Could the authors provide some details?

Response to comment 3 and the validation of the structural models using models with "probing data as constraints" is problematic. I insist this is a circular reasoning. I did not say that using the metrics of DSCI and/or AUROC is not correct. I meant you cannot use your own data to constraint a model and use the result as a reference to justify your data – that is circular reasoning. I don't think the other studies did this way. It's a common practice to use experimental structure (usually from a tertiary structure model) or computational structure (without probing data constraints) as reference. This is a severe problem affecting all comparisons.

Response to comment 10 is a confusing. The authors used DSCI quantitatively to define a specific set of regions with both high DSCI and high Δ DMS. The they claim that "This DSCI (or AUROC) is a very coarse metric". The question is, is it a metric or not? If "different structures can both have relatively high DSCI", what is the point to use DSCI for the evaluation of a structure?

Response to comment 14, I don't think it is necessary for the Ziv et al. 2020, Wan et al. 2021 and Yang et al. 2021 studies to be quantitative for the comparison. As the authors showed some agreement between their study and Ziv et al. 2020. But only one example is not enough.

Reviewer #2 (Remarks to the Author):

Great job addressing all of my comments and concerns. This is a very thorough revision!

Reviewer #3 (Remarks to the Author):

The authors had explained that existence of sgRNAs don't affect the conclusion of their study with two reasons: 1, 2/3 of the genome that can only come from full length molecules (including the FSE which

can only come from full length) have similar rate of alternative structures; 2, other work reports that the structure of the genomic and the sgRNAs are nearly identical (Yang et al. BioRxiv 2021) except near the 69 nt leader sequence (Sun et al).

According to the comparisons between the structures of sgRNAs from Yang et al. (BioRxiv 2021; Figure 4a) and genome RNA in this study (Fig 3), the alternative structures that exist in sgRNAs mainly located after 27500nt. It seems that there are few overlaps between the structures of sgRNA and the alternative structures of genome RNA, besides the 69nt leader sequence. As a result, it can't be ruled out that the structures in sgRNA may affect the analysis of the genome RNA, though this point don't affect the major conclusion of this study, and it should be discussed in the manuscript to make this study more rigorous. I have no other comments except this. In short, this revised manuscript has included more validation and cluster analysis with a new biological model. Considering the findings including the alternative structures in many regions that are consistent in two biological models, and a new model of FSE that was proved by reporter system, I think this study would make big contributions to exploring targets of viral replication or expression in RNA level.

Dear Dr. Ha,

Thanks to you and the reviewers for the valuable feedback, which has made our manuscript better. We appreciate the reviewers' assertions that our revisions were strong and that our study "would make big contributions to exploring targets of viral replication or expression in RNA level".

Importantly, in agreement with Reviewer 1, we have replaced the calculations in Figure 2 and used consensus structures for validation and comparison of data between our and other studies. We have also revised the text to avoid circular reasoning. Below we include a point-by-point response addressing the remaining issues.

Reviewer #1 (Remarks to the Author):

The revision has very nicely addressed many of my comments. The new data presented in the revision, e.g., the results from human cells, are very beautiful. The following are some remaining questions/concerns.

It is nice to see the new data in human cells and the agreement with the previous results in monkey cells. But I am a little worried to see the low agreements between all the structures, especially from the other studies. In particular, I would hope to see a decent agreement among those in vivo structures. PCC of 0.2~0.3 is way too low. How exactly the correlation is calculated? Could the authors provide some details?

We are very glad to hear we were able to nicely address many of your comments. We added more details on how the correlation was computed in the Methods (lines 850-66). Our data agrees very well between biological replicates (Pearson = 0.93, Fig. 1D) and decently with *in vitro* DMS data from Manfredonia et al (Pearson = 0.62, Fig. S6A).

The published datasets from groups using SHAPE derivatives have lower agreement with each other (Fig. S6B). We refrain from speculating why in the manuscript because it is difficult to determine the reasons – could be variations in the reagent or the library generation strategy. The original SHAPE molecules (used by Huston et al.) have limited permeability in mammalian cells, which stipulated the development of icSHAPE (Spitale et al., Nature, 2015). icSHAPE, used by Sun et al., is more cell permeable but has its own biases and may not report accurately on RNA structure (Busan et al., Biochemistry, 2019). More generally, SHAPE and its derivatives are much bulkier than DMS and are not specific to the Watson-Crick pairing positions that report directly on secondary structure. Instead, SHAPE chemicals react with the 2' OH of the ribose, which is an indirect measurement of base-pairing. Ours is the only study that used DMS in cells. DMS is the oldest and most established chemical for RNA structure mapping; it reports directly on base pairing (accessibility of N1A and N3C).

Response to comment 3 and the validation of the structural models using models with

"probing data as constraints" is problematic. I insist this is a circular reasoning. I did not say that using the metrics of DSCI and/or AUROC is not correct. I meant you cannot use your own data to constraint a model and use the result as a reference to justify your data – that is circular reasoning. I don't think the other studies did this way. It's a common practice to use experimental structure (usually from a tertiary structure model) or computational structure (without probing data constraints) as reference. This is a severe problem affecting all comparisons.

We apologize for not making this metric clearer. We are not using the AUROC score to validate or justify a given model but only to point out that some models are not in agreement with the actual data that was used to generate the model. This is an issue in prediction of RNA structure, because even though the data is used to constrain the prediction, depending on details of how the constrains were applied (e.g. the data could be weighted very weakly), the final model can be inconsistent with the underlying data. We therefore needed a metric based on which to discard models that are not a good fit to the data. We use the AUROC score only to discard a model or suspect alternative structures (Figure 3), simply because a low AUROC shows that the model is not a good solution to the raw data (new Supplementary Figure 10B). When used in this way, to assess the quality of a model that was generated with data constrains, a high AUROC cannot validate a model, it can only discard one. We have added new text (lines 286-296 and 959-72) and a new figure (Supplementary Figure 10B) where we demonstrate and clarify the usage of this metric.

Importantly, we entirely agree with the reviewer that using models constrained with our data to justify our data is a circular reasoning and this was not our intention. There are very few "gold standard" RNA structures for SARS-CoV-2 and it is difficult to validate without knowing the ground truth. However, because it is critical to omit all circular reasoning, we have replaced all the calculations of the quality assessment (Figure 2) to instead compare our and other models to the literature consensus structures for SARS-CoV-2 5' UTR that were generated prior to our work by multiple lines of evidence (updated Figure 2 and lines 169-89).

Response to comment 10 is a confusing. The authors used DSCI quantitatively to define a specific set of regions with both high DSCI and high Δ DMS. The they claim that "This DSCI (or AUROC) is a very coarse metric". The question is, is it a metric or not? If "different structures can both have relativity high DCSI", what is the point to use DCSI for the evaluation of a structure?

Indeed, we should have better clarified the usage of metric in the earlier manuscript versions. The point is simply for finding structures that are incorrect or likely have alternative conformations (because of low agreement with the data), rather than for validating a model. We now specifically added a figure (Supplementary Figure 10B) and text (lines 286-96) to demonstrate the application of this metric.

Response to comment 14, I don't think it is necessary for the Ziv et al. 2020, Wan et al. 2021 and Yang et al. 2021 studies to be quantitative for the comparison. As the authors showed some agreement between their study and Ziv et al. 2020. But only one example is not enough.

In their preprint, Yang and Wan et al. characterized SARS-CoV-2 RNA structures in Vero cells using Nanopore sequencing, proximity ligation, and SHAPE-MaP. Our genome-wide models agree on the well-validated structures in the 5' UTR (SL1 - SL7) and 3' UTR (BSL, P2, and s2m hairpin). Yang et al. also corroborate alternative stem 1 in the frameshifting stimulation element. In their Fig. 2b, they show 11 high-confidence structures comprising a total of 75 stems, and our genome-wide model agrees with 55 stems (73%).

There are a few studies now on the SARS-CoV-2 RNA structure but a detailed review of the agreements and discordances across all the literature is outside the scope of our current manuscript.

Reviewer #2 (Remarks to the Author):

Great job addressing all of my comments and concerns. This is a very thorough revision!

Reviewer #3 (Remarks to the Author):

The authors had explained that existence of sgRNAs don't affect the conclusion of their study with two reasons: 1, 2/3 of the genome that can only come from full length molecules (including the FSE which can only come from full length) have similar rate of alternative structures; 2, other work reports that the structure of the genomic and the sgRNAs are nearly identical (Yang et al. BioRxiv 2021) except near the 69 nt leader sequence (Sun et al).

According to the comparisons between the structures of sgRNAs from Yang et al. (BioRxiv 2021; Figure 4a) and genome RNA in this study (Fig 3), the alternative structures that exist in sgRNAs mainly located after 27500nt. It seems that there are few overlaps between the structures of sgRNA and the alternative structures of genome RNA, besides the 69nt leader sequence. As a result, it can't be ruled out that the structures in sgRNA may affect the analysis of the genome RNA, though this point don't affect the major conclusion of this study, and it should be discussed in the manuscript to make this study more rigorous.

We thank the reviewer for pointing out the potential effects of alternative structures in sgRNAs. We have added new text (lines 461-69) to discuss alternative structures in regions that Yang et al. found to be similar and different among different sgRNAs.

I have no other comments except this. In short, this revised manuscript has included

more validation and cluster analysis with a new biological model. Considering the findings including the alternative structures in many regions that are consistent in two biological models, and a new model of FSE that was proved by reporter system, I think this study would make big contributions to exploring targets of viral replication or expression in RNA level.

REVIEWERS' COMMENTS

Reviewer #1 (Remarks to the Author):

The revision has nicely addressed all my comments.